# THE LIPSCHITZ-VARIANCE-MARGIN TRADEOFF FOR ENHANCED RANDOMIZED SMOOTHING

**Blaise Delattre**[1,2]**, Alexandre Araujo**[3]**, Quentin Barthélemy**[1] **and Alexandre Allauzen**[2,4]

[1] Foxstream, Vaulx-en-Velin, France
[2] Miles Team, LAMSADE, Université Paris-Dauphine, PSL University, Paris, France
[3] ECE, New York University, NY, USA
[4] ESPCI PSL, Paris, France

## ABSTRACT

Real-life applications of deep neural networks are hindered by their unsteady predictions when faced with noisy inputs and adversarial attacks. The certified radius in this context is a crucial indicator of the robustness of models. However how to design an efficient classifier with an associated certified radius? Randomized smoothing provides a promising framework by relying on noise injection into the inputs to obtain a smoothed and robust classifier. In this paper, we first show that the variance introduced by the Monte-Carlo sampling in the randomized smoothing procedure estimate closely interacts with two other important properties of the classifier, *i.e.* its Lipschitz constant and margin. More precisely, our work emphasizes the dual impact of the Lipschitz constant of the base classifier, on both the smoothed classifier and the empirical variance. To increase the certified robust radius, we introduce a different way to convert logits to probability vectors for the base classifier to leverage the variance-margin trade-off. We leverage the use of Bernstein's concentration inequality along with enhanced Lipschitz bounds for randomized smoothing. Experimental results show a significant improvement in certified accuracy compared to current state-of-the-art methods. Our novel certification procedure allows us to use pre-trained models with randomized smoothing, effectively improving the current certification radius in a zero-shot manner.

## 1 INTRODUCTION

Deep neural networks are susceptible to adversarial attacks, which are small, carefully crafted perturbations that lead the model to make erroneous predictions (Szegedy et al., 2013). This vulnerability is a critical concern in applications requiring high reliability and safety, such as autonomous vehicles and medical diagnostics. Various defense mechanisms, including certified defenses like Lipschitz continuity (Cisse et al., 2017; Tsuzuku et al., 2018) and randomized smoothing (RS) (Cohen et al., 2019), have been proposed to mitigate these risks. Among the metrics used to evaluate these defenses, the certified robust radius serves as an important measure for quantifying model resilience against adversarial perturbations (Tsuzuku et al., 2018). The certified robust radius measures the amount of perturbation that can be added to an input $x$ while keeping the stability of the decision $y$, *i.e* the label in a classification task. This essentially acts as a certified measure of robustness for an individual input. Similarly, the prediction margin $M(f(x), y) := \max(0, f_y(x) - \max_{k \neq y} f_k(x))$ acts as an indicator of the confidence of the *base classifier* $f$ in assigning the label $y$ to the input $x$. A larger prediction margin correlates with increased confidence in the prediction, even if the input incurs some perturbations.

The concept of Lipschitz continuity augments this framework by introducing the Lipschitz constant which bounds the sensitivity of the *base classifier* to input perturbations. A smaller Lipschitz constant signifies that the function *base classifier* exhibits slower variations in its output with respect to changes in its input: $\|f(x+\tau) - f(x)\| \leq L(f)\|\tau\|$. Tsuzuku et al. (2018) gathers these elements to provide a bound on the certified robust radius that encompasses both the prediction margin and the Lipschitz constant. This combined measure controls the trade-off between the classifier's prediction margin and its sensitivity to input changes. Upon the introduction of RS, Li et al. (2018); Lecuyer

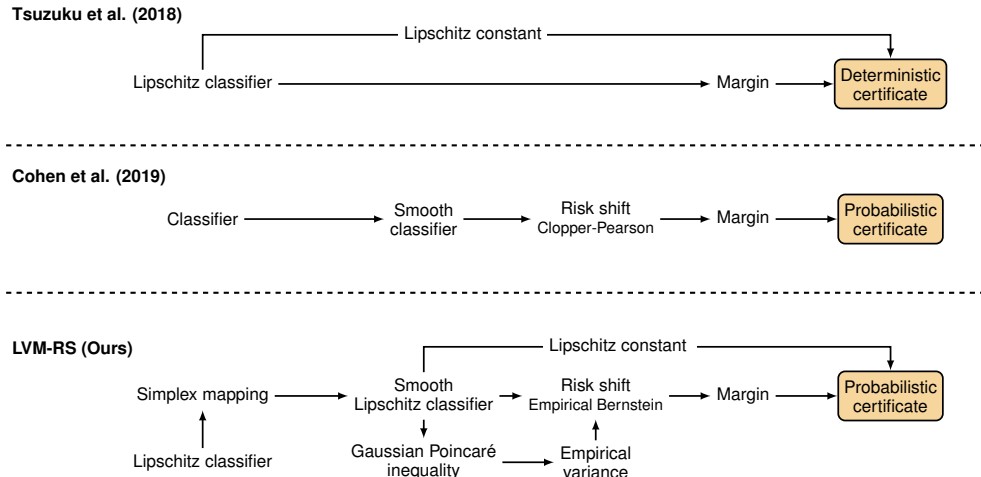

Figure 1: **First**, Tsuzuku et al. (2018) proposes a deterministic certificate starting from a Lipschitz *base subclassifier*, followed by margin calculation and radius binding. **Second**, Cohen et al. (2019) introduces a *base subclassifier* to create a *smoothed subclassifier*. The risk factor $\alpha$ is then estimated using the Clopper-Pearson interval to provide a probabilistic certificate. **Third**, our method (the *Lipschitz-Variance-Margin Randomized Smoothing* or LVM-RS) extends a smoothed classifier constructed with a Lipschitz base classifier composed with a map which transforms logit to probability vector in simplex. The regularization of the Lipschitz constant is motivated by the Gaussian-Poincaré inequality in Theorem 1. The empirical variance is applied to the Empirical Bernstein inequality in Proposition 2 to accommodate for the risk factor $\alpha$, in the same flavor as in Levine et al. (2020). The pipeline also ends with a probabilistic certificate, similar to the methodology used in Cohen et al. (2019)'s certified approach.

et al. (2019); Cohen et al. (2019); Salman et al. (2019); Levine et al. (2020) use the *smoothed classifier* obtained by convolving Gaussian density with the *base classifier*. Salman et al. (2019) proved that the *smoothed classifier* exhibits Lipschitz continuity which depends on the Gaussian variance. RS methods estimate a *smoothed classifier* by injecting noise on the input. The resulting procedure is then probabilistic and approximate inference is carried out with Monte-Carlo (MC) methods. To account for the randomness introduced through MC, one can use an $\alpha$-coverage confidence interval (*Clopper-Pearson*) as in (Cohen et al., 2019; Salman et al., 2019), or concentration inequality (*Hoeffding's inequality*) as in (Lecuyer et al., 2019; Levine et al., 2020) the probability to not predict the good label *i.e.* to control the risk induced by the randomness. A shift is thus necessary to lower-bound the prediction margin, thus yielding a more conservative but also more reliable estimate of the robust radius. In the last step of traditional classification, the decision usually relies on the $\arg\max$ function. This can be seen as a map of the network's output, putting all the probability mass on one corner of the simplex. However, in the context of RS, there is a detour: RS or the smoothed classifier conducts an $\arg\max$ map on the expectation of the $\arg\max$ applied to the base and deterministic classifier.

In this work, we revisit the whole decision process of RS to better leverage and disentangle the interplay of all these components. We propose to better leverage the margin-variance tradeoff with alternative simplex maps *i.e.* function to map logits to probability vector. More importantly, we investigate how Lipschitz regularity impacts randomized smoothing techniques, emphasizing its effects on the certified robust radius. The regularity of the *smoothed classifier* depends on the Lipschitz property of the *base classifier* and the variance of the Gaussian convolution which governs the induced level of smoothness. Therefore, our research in this domain encompasses following contributions:

- We use the *Gaussian-Poincaré's inequality* to explain the impact of the Lipschitz constant of the *base classifier* on MC variance, which ultimately affects its reliability, see Section 3.1. This motivates the use of the *Empirical Bernstein inequality* which integrates empirical variance to control risk $\alpha$, see Section 3.2.

- We introduce the $r$-simplex for RS procedure which allows for better-suited margins and Lipschitz constant of *smoothed classifier*, see Section 3.3. We establish a novel limit on the Lipschitz constant for the *smoothed classifier*, detailing its reliance on noise variance, simplex mass $r$, and the *base classifier*'s Lipschitz constant, whilst clarifying the connection between robustness certificates produced through Randomized Smoothing and the one from deterministic approaches as in Tsuzuku et al. (2018), see Section 3.4.

- We present the *Lipschitz-Variance-Margin Randomized Smoothing* (LVM-RS) procedure, presented in Figure 1, which balances MC variance and decision margin and controlled the MC empirical variance through the different simplex maps. This procedure demonstrates state-of-the-art results on the CIFAR-10 and ImageNet datasets, see Section 3.5.

## 2 BACKGROUND & RELATED WORK

The robustness of machine learning classifiers remains an active area of research, with various strategies being proposed and evaluated. In this section, we describe significant contributions in the domains of Lipschitz-based robust classifiers, randomized smoothing, and the role of margins in the robustness of classifiers.

### 2.1 NOTATION

Consider a $d$-dimensional input data point $x \in \mathcal{X} \subset \mathbb{R}^d$ and its associated label $y \in \mathcal{Y} = \{1, \ldots, c\}$, where $\mathcal{Y}$ encompasses $c$ distinct classes. The $(c-1)$-dimensional simplex is defined as $\Delta^{c-1} = \left\{ p \in \mathbb{R}^c \mid \mathbf{1}^\top p = 1, p \geq 0 \right\}$, and let $s : \mathbb{R}^c \mapsto \Delta^{c-1}$ denote the map onto this simplex. Usually, $s$ corresponds to the $\mathrm{softmax}$ or $\mathrm{hardmax}$ map. For a logit vector $z \in \mathbb{R}^c$, its map on $\Delta^{c-1}$ is denoted by $s(z)$. This map can use a temperature $t$ such that $s^t(z) := s(z/t)$. For instance, the $\mathrm{hardmax}$ corresponds for the component $k$ to $s_k(z) = \mathbb{1}_{\arg\max_i z_i = k}$ which puts all the mass on the maximum value coordinate. This map can be obtained through $\mathrm{softmax}$ with low temperature $t \to 0$. A function $f : \mathcal{X} \mapsto \mathbb{R}^c$ is designated as the subclassifier before the map $s$. The main classifier can be formulated as $F(x) := \arg\max_{k \in \mathcal{Y}} s_k(f(x))$, resulting in the predicted label $\hat{y} = F(x)$. While $F$ offers predictions for an input $x$, it doesn't convey the confidence level associated with these predictions.

The confidence level surrounding a classifier's decision boundary for a particular input $x$ is captured by the certified radius, denoted as $R(f, x)$. This radius represents the maximum permissible level of perturbation, $\epsilon$, that can be introduced to input $x$ without altering its classification output to remain consistent with its true label. A larger certified radius is indicative of a classifier's robustness against input perturbations. Its formal expression in the context of the $\ell_2$-norm is:

$$R(f, x) := \min_\epsilon \left\{ \epsilon > 0 \mid \exists \, \tau \in B_2(0, \epsilon), \, \arg\max_k f_k(x + \tau) \neq y \right\},$$

where $B_2(0, \epsilon) = \{\tau \in \mathbb{R}^d \mid \|\tau\|_2 \leq \epsilon\}$.

The local Lipschitz constant w.r.t the $\ell_2$-norm of a function $f$ over an open set $\mathcal{B}$ is defined as follows:

$$L(f, \mathcal{B}) = \sup_{\substack{x, x' \in \mathcal{B} \\ x \neq x'}} \frac{\|f(x) - f(x')\|_2}{\|x - x'\|_2} \, . \tag{1}$$

And if $L(f, \mathcal{B})$ exists and is finite, we say that $f$ is locally Lipschitz over $\mathcal{B}$. If $\mathcal{B} = \mathcal{X}$, we note $L(f, \mathcal{X}) = L(f)$ and call it Lipschitz constant.

### 2.2 LIPSCHITZ CONTINUITY IN CLASSIFIER DESIGN

The concept of Lipschitz continuity has been recognized for its intrinsic value in designing robust classifiers. By ensuring that a function possesses a bounded Lipschitz constant, it can be ascertained that small perturbations in the input don't result in large variations in the output.

**Proposition 1** (Tsuzuku et al. (2018)). *Given a Lipschitz continuous subclassifier $f$ for the $\ell_2$-norm, and given a perturbation level $\varepsilon > 0$, $x \in \mathcal{X}$, and $y \in \mathcal{Y}$ as the label of $x$. If the margin $M(f(x), y)$*

*at input $x$ meets the condition $M(f(x), y) > \sqrt{2}L(f)\varepsilon$, then for every $\tau$ such that $\|\tau\|_2 \leq \varepsilon$, we have $\arg\max_k f_k(x + \tau) = y$.*

Reworking this proposition, the certified radius for a subclassifier $f$ at $x, y$ can be expressed as:

$$R_1(f, x, y) := \frac{M(f(x), y)}{\sqrt{2}L(f)} . \tag{2}$$

This inherent property positions Lipschitz continuity as a strong defense mechanism against adversarial attacks. Recent efforts have focused on creating Lipschitz by design classifiers, incorporating Lipschitz constraints either during the training phase via regularization or through specific architectural designs Tsuzuku et al. (2018); Anil et al. (2019); Trockman & Kolter (2021); Singla & Feizi (2021b); Meunier et al. (2022); Araujo et al. (2023); Wang & Manchester (2023). Some works (Araujo et al., 2021; Singla & Feizi, 2021a; Delattre et al., 2023) provide soft Lipschitz constant regularization of individual layers. Other works consider local Lipschitz around input points, Huang et al. (2021); Muthukumar & Sulam (2023). However, there is a trade-off between the Lipschitz of the network and performance for the same level of margins, as depicted in Béthune et al. (2022, Appendix N). Instead of constraining the Lipschitz constant by design, methods commonly found in RS strategies, have shown better overall performance to procure certified robustness as they allow regular neural network architecture to be used.

### 2.3 RANDOMIZED SMOOTHING

First introduced by Lecuyer et al. (2019) and later developed in Li et al. (2018); Cohen et al. (2019); Salman et al. (2019) RS's central philosophy is to convolve the *base classifier* $F$ with a Gaussian distribution resulting in an *smoothed classifier* $\tilde{F}$ with increased robustness against adversarial inputs.

$$\tilde{F}(x) := \arg\max_k \mathbb{P}_{\delta \sim \mathcal{N}(0, \sigma^2 I)} (F(x + \delta) = k) = \arg\max_k \mathbb{E}_{\delta \sim \mathcal{N}(0, \sigma^2 I)}[\text{hardmax}_k(f(x))] .$$

For $s = \text{hardmax}$, we define the smoothed sub-classifier $\tilde{f} : \mathcal{X} \mapsto \mathbb{R}^c$ as follows:

$$\tilde{f}_k(x) := \mathbb{E}_{\delta \sim \mathcal{N}(0, \sigma^2 I)} [s_k (f(x + \delta))] ,$$

and, in this article, $\tilde{F}$ is generalized to any simplex map $s$:

$$\tilde{F}_s(x) = \arg\max_k \mathbb{E}_{\delta \sim \mathcal{N}(0, \sigma^2 I)} [s_k (f(x + \delta))] = \arg\max_k \tilde{f}_k(x) ,$$

where $s$ corresponds in this context to the $\text{hardmax}$ map on the simplex, applied to the *base sub-classifier* $f$ through the RS process (Cohen et al., 2019; Salman et al., 2019) [1]. We note $p = \tilde{f}(x)$ and suppose $p$ is sorted in decreasing order. Certified radius makes the mapping $\Phi^{-1} \circ \tilde{f}_k$ intervene, where $\Phi^{-1}$ represents the quantile function of the Gaussian distribution. We suppose here that the classifier $\tilde{F}_s$ gives the good answer, the certified radius writes as:

$$R_2(p) = \frac{\sigma}{2} \left( \Phi^{-1}(p_1) - \Phi^{-1}(p_2) \right) . \tag{3}$$

In most RS approaches, the bound on $p_2 \leq 1 - p_1$ is used, it degrades the certified radius especially when the smoothing noise $\sigma$ is high, see Appendix B. RS uses MC method to estimate $p$ by $\hat{p} = \frac{1}{n} \sum_{i=1}^n s(f(x + \delta_i))$, where $\delta_i$ are sampled from a Gaussian distribution. As the RS method is probabilistic, there is a risk $\alpha$ that the method returns the wrong answer (Cohen et al., 2019). Following Levine et al. (2020), a confidence interval bound or concentration inequality is used to provide a $\text{shift}(S_n(\hat{p}_k), \alpha, n)$ such that:

$$\mathbb{P}\left( \bigcap_{k=1}^c (\hat{p}_k - \text{shift}(S_n(\hat{p}_k), \alpha, n) \leq p_k \leq \hat{p}_k + \text{shift}(S_n(\hat{p}_k), \alpha, n)) \right) \leq 1 - \alpha , \tag{4}$$

where $S_n$ is the sample variance. Then, the risk-corrected probabilities are obtained

$$\bar{p} = (\hat{p}_1 - \text{shift}(S_n(\hat{p}_k), \alpha, n), \ldots, \hat{p}_2 + \text{shift}(S_n(\hat{p}_2), \alpha, n), \ldots, \hat{p}_c + \text{shift}(S_n(\hat{p}_c), \alpha, n)) ,$$

---

[1]Levine et al. (2020) do not use a simplex map but normalized outputs in $[0, 1]$ for another task.

supposing that the $(\bar{p}_k)_{k=2}^c$ are sorted in decreasing order, $\bar{p}_1$ and $\bar{p}_2$ are used to compute the risk corrected radius $R_2(\bar{p})$ which holds with probability $1 - \alpha$.

The drawback of RS is the sampling cost, as MC error reduces with $\frac{1}{\sqrt{n}}$ with $n$ the number of samples. To tackle this issue the work of Horváth et al. (2022) leverages an ensemble of classifiers to reduce the variance and proposes an adaptive sampling procedure to verify whether a target-certified radius is reached or not. In this work, we also aim to reduce the variance of the MC sampling.

### 2.4 MARGINS AND CLASSIFIER ROBUSTNESS

The margin, often described as the distance between the decision boundary and the nearest data instance, serves as a central component of classifier robustness. Larger margins are generally associated with better generalization capabilities, a main principle behind algorithms such as support vector machines. In the context of adversarial robustness, margins play a critical role, with several studies highlighting the relationship between margins and resilience against adversarial perturbations. Efforts to optimize for larger margins, combined with other robustness-enhancing strategies, have shown promise in strengthening classifier defenses. The work of Béthune et al. (2022) explores the connection between margin maximization and Lipschitz continuity, and shows how both notions implement a tradeoff between accuracy and robustness.

While RS and Lipschitz continuity by design have been studied in their distinct capacities, recent research suggests an inherent synergy between them and the key role of margins. Our work focuses on the connection between RS and Lipschitz continuity to produce greater robustness.

## 3 THE LIPSCHITZ-VARIANCE-MARGIN TRADEOFF

In this section, we explain why the control of the Lipschitz constant in the subclassifier is crucial to reduce the MC variance of RS. By applying *Bernstein's inequality*, we can decrease variance to improve the control of risk $\alpha$, and to further reduce variance, we employ a new simplex mapping on $f$, thus defining the LVM-RS procedure. Furthermore, we establish new bounds on the Lipschitz constant of the *smoothed classifier* w.r.t the Lipschitz constant of the *subclassifier*. In the following, we defer all proofs in the appendix.

### 3.1 LOW LIPSCHITZ FOR LOW VARIANCE

The concept of Lipschitz continuity plays an important role in the sampling process, which is crucial for obtaining an accurate estimation of the smoothed classifier $\tilde{f}$. Specifically, by minimizing the local Lipschitz constant of a subclassifier $s \circ f$, one can reduce its variance. The following theorem illustrates this relationship for any $\sigma$.

**Theorem 1** (Gaussian Poincaré inequality (Boucheron et al., 2013)). *Let $Z = (Z_1, \ldots, Z_n)$ represent a vector of i.i.d Gaussian random variables with variance $\sigma^2$. For any continuously differentiable function $h : \mathbb{R}^n \to \mathbb{R}$, the variance is given by:*

$$\mathbb{V}[h(Z)] \leq \sigma^2 \, \mathbb{E}\left[\|\nabla h(Z)\|^2\right] \; .$$

We use the latter theorem to immediately derive:

**Corollary 1.** *With same hypothesis as Theorem 1, if $h$ exhibits Lipschitz continuity, we have that:*

$$\mathbb{V}[h(Z)] \leq \sigma^2 \, L(h)^2 \; .$$

Applying the above corollary to the classifiers $\{s_k \circ f\}$ which can be considered differentiable almost everywhere, it is evident that constraining the Lipschitz constant, $L(s_k \circ f)$, leads to a diminished variance for $s_k \circ f$. This, in turn, results in a more precise estimation of $\mathbb{E}[s(f(x + \delta))]$, as captured by $\frac{1}{n} \sum_{i=1}^{n} s(f(x + \delta_i))$. Lowering the local Lipschitz constraints can significantly attenuate the variance and improve the certification results, but it can be too restrictive and cause a drop in performance. To enforce low Lipschitz and reduce performance loss, Cohen et al. (2019) proposed the injection of Gaussian noise during training, Salman et al. (2019) introduced $\mathrm{SmoothAdv}$, which involves adversarial training of the smoothed classifier $\tilde{f}$, to reduce its local Lipschitz constant. The

work of Pal & Sulam (2023) studied how the noisy training on the *sub classifier* affects the performance and robustness of the *smoothed classifier*. Other noteworthy methods include those by Salman et al. (2020); Carlini et al. (2023), which combine a conventional classifier with a denoiser diffusion model, ensuring that the resulting architecture remains invariant to Gaussian noise, thereby giving Lipschitz continuity to the classifier and preserved performance.

## 3.2 Statistical Risk Management for Low Variance

To leverage low variance, one would need a $\alpha$ confidence interval or an appropriate concentration inequality in which variance plays a significant role.

The *Clopper-Pearson* binomial tailored confidence interval can be used to give an exact $\alpha$ coverage to determine $\mathrm{shift}$ defined in Eq. (4), Cohen et al. (2019); Carlini et al. (2023). It is paired with $\mathrm{hardmax}$ simplex map which generates a series of Bernoulli trials during MC sampling.

Lecuyer et al. (2019) and Levine et al. (2020) smoothed scalar outputs between within [0, 1] and cannot use such interval, we use the same procedure as those works. They rely upon *Hoeffding's inequality*, which also gives exact an $\alpha$ coverage. Another interesting inequality, similar to the *Gaussian-Poincaré*, is the sub-Gaussian inequality involving the Lipschtiz constant of $s_k \circ f$, Massart (2007). The issue here is that computing $L(s_k \circ f)$ is NP-hard for common neural networks and the Lipschitz constant as a bound can overestimate the actual empirical variance. However, those inequalities have some limitations because they do not account for empirical variance. This is why we suggest employing the *Empirical Bernstein's inequality* when the variance is low to manage the risk, $\alpha$, which does factor in the observed empirical variance, it has been mentioned by Lecuyer et al. (2019).

**Proposition 2** (Empirical Bernstein's inequality (Maurer & Pontil, 2009)). *Let $Z_0, Z_1, \ldots, Z_n$ be i.i.d random variables with values between 0 and 1. The risk level is denoted as $\alpha \in [0, 1]$. Then with probability at least $1 - \alpha$ in vector $Z = (Z_1, \ldots, Z_n)$, we have*

$$\mathbb{E}Z_0 - \frac{1}{n}\sum_{i=1}^{n} Z_i \leq \sqrt{\frac{2S_n(Z)\log(2/\alpha)}{n}} + \frac{7\log(2/\alpha)}{3(n-1)} := \mathrm{shift}(S_n(Z), \alpha, n) .$$

*Here, $S_n(Z)$ represents the sample variance $\frac{1}{n(n-1)}\sum_{1 \leq i < j \leq n}(Z_i - Z_j)^2$. Note that the bound is symmetric about $\mathbb{E}Z_0$.*

In our case, we use this inequality with $Z_i = s_k(f(x + \delta_i))$. This inequality offers the flexibility to smooth various simplex maps $s$ and potentially select one better equipped than $\mathrm{hardmax}$ to address the margin-variance tradeoff. See Fig. 2, for a comparison between Hoeffding's and Bernstein's inequalities.

## 3.3 High margin and Low Variance by optimal mapping on r-simplex

As all the mass is put on one class, $\mathrm{hardmax}$ map gives maximal margins on the simplex. However, it is not Lipschitz continuous, and it can increase variance, as illustrated in Example C.1. Conversely, $\mathrm{softmax}$ map compresses the margins between classes but its $1-$Lipschitz continuity prohibits variance amplifications. Martins & Astudillo (2016) introduced a novel simplex mapping, 1-Lipschitz and producing margins larger than $\mathrm{softmax}$ but lower than $\mathrm{hardmax}$: $\mathrm{sparsemax}(z) = \arg\min_{p \in \Delta^{c-1}}\|p - z\|_2^2$. This map promotes sparse values in $\Delta^{c-1}$ compared to the softmax.

In this work, we introduce the $r$-simplex $\Delta_r^{c-1} = \{p \in \mathbb{R}^c \mid \mathbf{1}^\top p = r, p \geq 0\}$ a simplex with total mass $r$ and the generalized $\mathrm{sparsemax}$ which is a 1-Lipschitz mapping towards $\Delta_r^{c-1}$, following the same proof as in (Laha et al., 2018, Appendix A.5). This new mapping is described in Algo. 1. For $r_1 \leq r_2$, when most of the logit vectors $f(x + \delta_i)$ are bounded by $r_1$, the mapping to the simplex $\Delta_{r_2}^{c-1}$ with 1-Lipschitz simplex mapping is not going to increase the margin associated to vectors $s_{r_2}(f(x+\delta_i))$. In this case, it is better to map to the simplex $\Delta_{r_1}^{c-1}$ of lower mass and enjoy a tighter Lipschitz constant on $\tilde{f}$ or $\Phi^{-1} \circ \tilde{f}_k$. Conversely, when $r_2 \geq r_1$, one can benefit from larger margins on $s_{r_2}(f(x+\delta_i))$ in comparison to margins on $s_{r_1}(f(x+\delta_i))$.

In addition, we add temperature to simplex mappings: we note $s^{t,r}$ the $r$-simplex map from $\mathbb{R}^c$ to $\Delta_r^{c-1}$ for a temperature $t$. Adjusting the temperature provides a means to interpolate between softmax and hardmax, or between sparsemax and hardmax. Tuning the temperature allows us to find an optimized simplex map to answer the variance-margin trade-off, as illustrated in Fig. 3.

## 3.4 NEW OPTIMAL LIPSCHITZ BOUNDS FOR RS

We derive enhanced bounds on the Lipschitz constant of the smoothed classifier $\tilde{f}$ with the additional assumption that $s_k^r \circ f$ or $s^r \circ f$ themselves are Lipschitz continuous. Note that one way to have $s^r \circ f$ or $s_k^r \circ f$ Lipschitz continuous is to have $s$ Lipschitz continuous as well. This is not the case of hardmax simplex map, whereas sparsemax is ideal as it is 1-Lipschitz continuous and conserves margin as long as the latter is inferior to simplex mass $r$.

**Theorem 2.** *Let $f : \mathcal{X} \subset \mathbb{R}^d \mapsto \mathbb{R}^c$ a subclassifier and $\tilde{f}(x) = \mathbb{E}_{\delta \sim \mathcal{N}(0,\sigma^2 I)}[s^r(f(x + \delta))]$ the associated smoothed classifier. Suppose that $f$ is element-wise Lipschitz continuous, then*

$$L(\tilde{f}_k) \leq L(s_k^r \circ f) \operatorname{erf}\left(\frac{r}{2^{\frac{3}{2}} L(s_k^r \circ f)\sigma}\right) \leq \min\left\{\frac{r}{\sqrt{2\pi\sigma^2}}, L(s_k^r \circ f)\right\} . \tag{5}$$

*Suppose that $f$ is Lipschitz continuous, then*

$$L(\tilde{f}) \leq L(s^r \circ f) \operatorname{erf}\left(\frac{r}{2L(s^r \circ f)\sigma}\right) \leq \min\left\{\frac{r}{\sqrt{\pi\sigma^2}}, L(s^r \circ f)\right\} . \tag{6}$$

It is noteworthy that Eq. (5) enhances the bound on $L(\tilde{f}_k)$ originally derived in Lemma 1 of Salman et al. (2019) for $r = 1$ by a factor of 2. This refinement on the bound was possible by supposing Lipschitz continuity on the *base classifier* $f$. Note that its Lipschitz constant can be arbitrarily high, so this assumption is quite light: the Lipschitz constant does not play into the derived bound. These improved bounds can be seamlessly incorporated into subsequent works, such as Pautov et al. (2022); Franco et al. (2023); Chen et al. (2024).

We observe that randomized smoothing and Lipschitz continuity exhibit a cross-effect on the Lipschitz constant of $\tilde{f}$. We focus on an intermediate regime defined by a specific $\sigma$ and $L(s \circ f)$, where these effects interact in a manner that is mutually beneficial, exceeding the individual impacts of randomized smoothing or Lipschitz continuity alone.

**Proposition 3.** *The optimal value $\sigma^*$ that maximizes the gap between the bounds of Eq. (5) is:*

$$\sigma^* = \frac{r}{L(s_k^r \circ f)\sqrt{2\pi}} \quad \text{giving} \quad L(\tilde{f}_k) \leq \operatorname{erf}(\sqrt{\pi}/2) \, L(s_k^r \circ f) \lesssim 0.79 \, L(s_k^r \circ f) . \tag{7}$$

*Similarly, for Eq. (6):*

$$\sigma^* = \frac{r}{L(s^r \circ f)\sqrt{\pi}} \quad \text{giving} \quad L(\tilde{f}) \leq \operatorname{erf}(\sqrt{\pi}/2) \, L(s^r \circ f) \lesssim 0.79 \, L(s^r \circ f) . \tag{8}$$

For this choice of $\sigma^*$, $L(s_k^r \circ f)$ equals the RS bound (and is exactly the deterministic Lipschitz constant). Consequently, the combined use of Lipschitz continuity and randomized smoothing reduces the Lipschitz constant bound of $\tilde{f}$ by at most 21%. In our framework, given either a Lipschitz constant (or $\sigma^2$), one can select the complementary $\sigma^2$ (or Lipschitz constant) to maximize the synergistic effects of randomized smoothing and inherent Lipschitz continuity. For this optimal choice, we obtain a certificate Eq. (2) that is approximately 26% larger than the maximum certification given by RS or Lipschitz continuity alone.

All previous bounds are derived on the Lipschitz constant of $\tilde{f}$ which can be used for radius $R_1$. With regards to $R_2$, we have the following result on the local Lipschitz constant of $\Phi^{-1} \circ \tilde{f}_k$ :

**Theorem 3.** *Let $\tilde{f} : \mathbb{R}^d \mapsto \Delta_r^{c-1}$ be the smoothed classifier, for an input $x \in \mathcal{X}$ and $\mathcal{B} = B_2(x, \epsilon)$, the local Lipschitz constant of $\Phi^{-1} \circ \tilde{f}_k$ is bounded by:*

$$L\left(\Phi^{-1} \circ \tilde{f}_k, \, \mathcal{B}\right) \leq \frac{r}{\sigma} \max_{p \in B_2\left(\tilde{f}(x), \, \epsilon L(\tilde{f})\right)} \left\{\exp\left(-\frac{1}{2}\left(\Phi^{-1}(p/r)^2 - \Phi^{-1}(p)^2\right)\right)\right\} .$$

This result gives a local Lipschitz constant on a $\epsilon$ ball around $x$, in the same flavor as in (Muthukumar & Sulam, 2023).

## 3.5 LVM-RS INFERENCE PROCEDURE

Given a trained base subclassifier $f$, we choose a simplex mapping $s$ from a set of simplex map $\mathcal{S}$ and a temperature $t \in [t_{\text{lower}}, t_{\text{upper}}]$, defining an ensemble of classifiers $\{\tilde{F}_{s^t}\}$:

$$\tilde{F}_{s^t}(x) = \arg\max_k \mathbb{E}_{\delta \sim \mathcal{N}(0, \sigma^2 I)} \left[ s_k^t(f(x + \delta)) \right] = \arg\max_k \tilde{f}_{s^t}(x) .$$

First we generate a test sample of size $n_0$, we obtain estimates $\hat{p}_{s^t}$, using *Bernstein's inequality* we obtain the risk corrected $\bar{p}_{s^t}$ and finally the risk corrected certified radius $R_2(\bar{p}_{s^t})$. We choose $(s_*, t_*) = \arg\max_{s,t} R_2(\bar{p}_{s^t})$ to maximize the certified radius. Then a sampling of size $n$ is performed and we evaluate an MC estimate of $\tilde{f}_{s_*^{t_*}}(x)$ which gives $\hat{p}^*$ and associated risk corrected $\bar{p}^*$. We return prediction $\arg\max_k \bar{p}_k^*$ and associated certified radius $R_2(\bar{p}^*)$. Our approach summed up in Algo. 2, addresses the trade-off between maximizing margins and reducing variances. The procedure SampleScores generates scores $f(x + \delta_i)$ for $\delta_i$ samples from $\mathcal{N}(0, \sigma^2 I)$. This stands in contrast to methods like hardmax, which maximize margin at the cost of increased variance, and others like sparsemax and softmax, which prioritize reduced variance over margin maximization.

## 4 EXPERIMENTS

In addition to the two experiments below, we conduct an ablation study in Appendix F.

### 4.1 CERTIFIED ACCURACY WITH IMPROVED RS LIPSCHITZ BOUND

Table 1: Certified accuracy on CIFAR-10 for different levels of perturbation $\epsilon$, for RS, Lipschitz deterministic, and ours. The risk is taken as $\alpha = 1e-3$ and the number of samples $n = 10^4$.

| Methods | Certified accuracy ($\varepsilon$) | | | | | | Average time (s) |
|---|---|---|---|---|---|---|---|
| | 0.14 | 0.19 | 0.25 | 0.28 | 0.42 | 0.5 | |
| Lipschitz deterministic | 40 | 33.57 | 27.18 | 24.59 | 13.65 | 9.15 | **0.004** |
| Randomized smoothing | 47.9 | 31.99 | 28.17 | 27.86 | 6.42 | 0.0 | 0.9 |
| **RS with new bound** | **52.56** | **46.17** | **39.09** | **35.08** | **21.9** | **13.53** | 0.9 |

To illustrate the gain of having a Lipschitz bound of *smoothed classifier* which includes information on the Lipschitz constant of *sub classifier*, simplex mass $r$ and variance $\sigma^2$, we compare certified accuracies on the same by design 5-Lipschitz backbone Sandwhich Small from (Wang & Manchester, 2023), trained with noise injection $\sigma = 0.4$ and using the same certified robust radius $R_1$ in Eq. (2). We choose for the smoothing variance $\sigma^* = \frac{r}{L(f)\sqrt{\pi}}$ as explained in Section 3.4. Remark that the variance used for noise injection to train the 5-Lipschitz *sub classifier* is close to $\sigma^*$ to mitigate a drop in performance on the *smoothed classifier*.

**Impact of Lipschitz constant** We consider the three procedures: Lipschitz deterministic (using bound on $L(f)$), the RS (using bound on $L(\tilde{f})$), and our approach (using bound on $L(\tilde{f})$). For ours and RS, we fix $r = 3$. Results are displayed in Table 1. We see that our procedure gives better-certified accuracies than RS and Lipschitz deterministic taken alone, indeed both methodologies provide the same Lipschitz constant for $\tilde{f}$ and $f$ respectively, whereas our method provides an inferior Lipschitz bound on $L(\tilde{f})$. Note that better results from the random procedure should not be directly construed as an intrinsic superiority over the deterministic one, as the element of randomness introduces variability that must be accounted for in the evaluation and large sampling computational cost. However, it gives a perspective over the performance of the theoretical Lipschitz *smoothed classifier* $\tilde{f}$.

**Impact of simplex mass** We note that to reduce the Lipschitz constant, one can not only increase the smoothing noise $\sigma$ as done traditionally in RS but also reduce $r$ the total mass of the simplex. We plot the evolutions of certified accuracies for different simplex masses in Fig. 4 with the same setting as above. We see that classical RS setting *i.e.* $r = 1.0$ is one particular choice of robustness profile among many.

Table 2: Best certified accuracies across $\sigma \in \{0.25, 0.5, 1.0\}$ for different levels of perturbation $\epsilon$, on CIFAR-10, for $n = 10^5$ samples and risk $\alpha = 1\mathrm{e}{-3}$.

| Methods | Best certified accuracy ($\varepsilon$) | | | | | Average time (s) |
|---|---|---|---|---|---|---|
| | 0.0 | 0.25 | 0.5 | 0.75 | 1.0 | |
| Carlini et al. (2023) | 86.72 | 74.41 | 58.25 | 40.96 | 29.91 | 7.10 |
| **LVM-RS (ours)** | **88.49** | **76.21** | **60.22** | **43.76** | **32.35** | 7.11 |

Table 3: Best certified accuracies across $\sigma \in \{0.25, 0.5, 1.0\}$ for different levels of perturbation $\epsilon$, on ImageNet, for $n = 10^4$ samples and risk $\alpha = 1\mathrm{e}{-3}$.

| Methods | Best certified accuracy ($\varepsilon$) | | | | | | Average time (s) |
|---|---|---|---|---|---|---|---|
| | 0.0 | 0.5 | 1.0 | 1.5 | 2 | 3 | |
| Carlini et al. (2023) | 79.88 | 69.57 | 51.55 | **36.04** | 25.53 | 14.01 | 6.46 |
| **LVM-RS (ours)** | **80.66** | **69.84** | **53.85** | **36.04** | **27.43** | **14.31** | 7.03 |

## 4.2 CERTIFIED ACCURACY WITH LVM-RS

In this experiment, we empirically validate the efficacy of our proposed inference procedure presented in Algo. 2, highlighting its capability to improve randomized smoothing and achieve certified accuracy. Central to our approach is the leveraging of the variance-margin tradeoff, which as we demonstrate, yields state-of-the-art RS results. We further showcase how the procedure enhances the off-the-shelf state-of-the-art baseline model of Carlini et al. (2023), which utilizes a vision transformer coupled with a denoiser for randomized smoothing. We use 50 temperatures ranging from $t_{\mathrm{lower}} = 0.01, t_{\mathrm{upper}} = 50$, and simplex maps $\mathcal{S} = \{\mathrm{sparsemax}, \mathrm{softmax}, \mathrm{hardmax}\}$. The baseline consists of the state-of-the-art top performative model of Carlini et al. (2023) which does smoothing of $\mathrm{hardmax}$ of *base classifier* and uses the *Pearson-Clopper* confidence interval to control the risk $\alpha$.

To compare the baseline with our method, certified accuracies are computed with $R_2$ in the function of the level of perturbations $\epsilon$, for different noise levels $\sigma = \{0.25, 0.5, 1\}$. Results are presented in Figure 5 for CIFAR-10 and in Figure 6 for ImageNet. We see that our method increases results, especially in the case of high $\sigma$, in the case of $\sigma \in \{0.5, 1.0\}$ the overall certified accuracy curve in the function of $\epsilon$ the maximum perturbation is lifted towards higher accuracies. Results are presented in Table 2 for CIFAR-10 and in Table 3 for ImageNet. Computation was performed on GPU V100, reported average time is the computational cost of one input $x$ proceeds by RS and LVM-RS, we see that the computation gap between the two methods is narrow for CIFAR10 but is a bit wider for ImageNet. Detailed results are presented in Appendix G.

## 5 CONCLUSION

In this paper, we demonstrate a significant connection between the variance of randomized smoothing and two critical properties of the *subclassifier*: its Lipschitz constant and its margin. We highlight the influence of the Lipschitz constant on both the *smoothed classifier* and the empirical variance. To improve the certified robust radius, new simplex of mass $r$ and simplex map are introduced for the *subclassifier*, which optimally manages the Lipschitz-variance-margin trade-off. Along with this, we incorporate an advanced Lipschitz bound for the RS, resulting in improved certified accuracy compared to the prevailing methods. In addition, our new certification procedure facilitates the use of pre-trained models in conjunction with randomized smoothing, leading to a direct improvement in the current certification radius. In future research, we plan to integrate LVM-RS with margin maximization strategies and explore the choice of the simplex mass $r$.

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

## A    ACKNOWLEDGMENTS

This work was performed using HPC resources from GENCI- IDRIS (Grant 2023-AD011014214) and funded by the French National Research Agency (ANR SPEED-20-CE23-0025). This work received funding from the French Government via the program France 2030 ANR-23-PEIA-0008, SHARP. We thank Yann Chevaleyre, Muni Sreenivas Pydi, Florian Le Bronnec, and Sylvain Delattre for their precious help on the Proof D.1.

## B    RELATION BETWEEN CERTIFIED RADII

Using the fact that $p_2 \leq 1 - p_1$ into Eq. (3), Cohen et al. (2019, Section 3.2.2) motivate the use of certificate $R_3$ for statistical simplicity and saying that $f(x + \delta)$ puts most of its weight on the top class:

$$R_3(p) = \sigma\Phi^{-1}(p_1) \leq R_2(p) . \tag{9}$$

For the Carlini et al. (2023) architecture, it is not an optimal choice: in the context of high variance $\sigma^2$, the distribution of $\tilde{f}(x)$ tends to be closer to a uniform distribution. Thus, the difference between radii $R_2$ and $R_3$ increases, as shown in Table 4. This effect has been noted in (Voráček & Hein, 2023).

Table 4:   Comparison of two certified radii $R_2$ and $R_3$, and Total Variation distance to Uniform distribution (TVU), for different values of $\sigma$. We took a subset of images from the ImageNet test set with $n = 10^4$, $\alpha = 0.001$, and used Empirical Bernstein's inequality. If the effect is not visible for $\sigma < 0.25$, we see that as the TVU decreases, the difference between the two radii increases as well.

| $\sigma$ | TVU | $R_2(\bar{p})$ | $R_3(\bar{p})$ | $R_2(\bar{p}) - R_3(\bar{p})$ |
|---|---|---|---|---|
| 0.25 | 0.998 | **5.89** | **5.89** | 0.00 |
| 0.30 | 0.996 | **4.68** | 4.48 | 0.20 |
| 0.35 | 0.989 | **3.68** | 3.21 | 0.47 |
| 0.40 | 0.986 | **2.49** | 1.97 | 0.52 |
| 0.50 | 0.976 | **1.28** | 0.59 | 0.69 |
| 0.60 | 0.95 | **0.56** | 0.00 | 0.56 |

# C    SIMPLEX MAPPING

## C.1    EXAMPLE ON hardmax

**Example.** *For a random variable $X$ defined over the interval $[0, 1]$, with $\mathbb{E}[X] = \frac{1}{2}$ and a "small" variance $\mathrm{Var}[X] = \sigma^2 < \sigma^2_{\max} = \frac{1}{4}$, we define a new random variable as $Y = s(X) = \mathbb{1}_{X > \frac{1}{2}}$. Then $\mathrm{Var}[Y] = \sigma^2_{\max}$ will be much higher than $\mathrm{Var}[X]$ when $\sigma^2$ is significantly smaller than $\frac{1}{4}$.*

*Proof.* Let us compute $\mathbb{E}[Y]$, since $Y$ is an indicator random variable, we have: $\mathbb{E}[Y] = \mathbb{P}(Y = 1) = \mathbb{P}(X > \frac{1}{2})$. Given $X$ is symmetric around 0.5, we find $\mathbb{E}[Y] = 0.5$. The variance of $Y$ is given by: $\mathbb{V}[Y] = \mathbb{E}[Y^2] - (\mathbb{E}[Y])^2$. Since $Y$ is an indicator variable, $Y^2 = Y$, and therefore $\mathbb{E}[Y^2] = \mathbb{E}[Y] = 0.5$. Thus, we have: $\mathbb{V}[Y] = 0.5 - 0.25 = 0.25$. To claim that $\mathbb{V}[Y]$ has a much higher variance than $\mathbb{V}[X]$, we need to compare 0.25 to $\sigma^2$. The statement would be true if $\sigma^2$ is substantially smaller than 0.25. Given that $X$ is defined over $[0, 1]$ and its mean is 0.5, the variance of $X$ could range between 0 and 0.25. Therefore, unless $X$ has a variance near this upper bound, $\mathbb{V}[Y]$ will indeed be much larger. $\qquad\square$

## C.2    ALGORITHM OF GENERALIZED SPARSEMAX

---

**Algorithm 1** generalized_sparsemax($z$, $r$)

---

1: Sort $z$ in decreasing order $z_1 \geq \cdots \geq z_c$
2: Find $\kappa(z)$ such that
$$\kappa(z) = \max_{k=1\dots c} \left\{ k \,\middle|\, r + kz_k > \sum_{j \leq k} z_j \right\}$$
3: Define
$$\rho(z) = \frac{\left( \sum_{j \leq \kappa(z)} z_j \right) - r}{\kappa(z)}$$
4: **return** $p$ such that $p_i = \max(z_i - \rho(z), 0)$

---

# D  PROOFS FOR LIPSCHITZ BOUNDS FOR RS

## D.1  PROOF OF THEOREM 2

For both parts of the proof, we are going to use the following lemmas.

**Lemma 1.** *(Stein's lemma (Stein, 1981, Lemma 2))*
*Let $\sigma > 0$, let $h : \mathbb{R}^d \mapsto \mathbb{R}$ be measurable, and let $\tilde{h}(x) = \mathbb{E}_{\delta \sim \mathcal{N}(0,\sigma^2 I)}[h(x + \delta)]$. Then $\tilde{h}$ is differentiable, and moreover,*

$$\nabla \tilde{h}(x) = \frac{1}{\sigma^2} \mathbb{E}_{\delta \sim \mathcal{N}(0,\sigma^2 I)}[\delta h(x + \delta)] \,.$$

Stein's lemma can be easily extended to $h : \mathbb{R}^d \mapsto \mathbb{R}^c$. We note $\frac{\partial}{\partial x}\tilde{h}(x)$ the Jacobian matrix of $\tilde{h}(x)$.

**Lemma 2.** *Let $\sigma > 0$, let $h : \mathbb{R}^d \mapsto \mathbb{R}^c$ be measurable, and let $\tilde{h}(x) = \mathbb{E}_{\delta \sim \mathcal{N}(0,\sigma^2 I)}[h(x + \delta)]$. Then $\tilde{h}$ is differentiable, and moreover,*

$$\frac{\partial \tilde{h}(x)}{\partial x} = \frac{1}{\sigma^2} \mathbb{E}_{\delta \sim \mathcal{N}(0,\sigma^2 I)}[\delta h(x + \delta)^\top] \,.$$

*Proof.* We are not going to prove differentiability as it is the same argument as in the proof of Lemma 1, see (Stein, 1981).

$$
\begin{aligned}
\frac{\partial}{\partial x}\tilde{h}(x) &= \frac{\partial}{\partial x}\left(\frac{1}{(2\pi\sigma^2)^{d/2}}\int_{\mathbb{R}^d} h(t)\exp\left(-\frac{1}{2\sigma^2}\|x - t\|^2\right)dt\right) \\
&= \frac{1}{(2\pi\sigma^2)^{d/2}}\int_{\mathbb{R}^d}\frac{\partial}{\partial x}\left(\exp\left(-\frac{1}{2\sigma^2}\|x - t\|^2\right)\right)h(t)^\top dt \\
&= \frac{1}{(2\pi\sigma^2)^{d/2}}\int_{\mathbb{R}^d}\frac{\partial}{\partial x}\left(\exp\left(-\frac{1}{2\sigma^2}\|x - t\|^2\right)\right)h(t)^\top dt \\
&= \frac{1}{(2\pi\sigma^2)^{d/2}}\int_{\mathbb{R}^d}\frac{t - x}{\sigma^2}\exp\left(-\frac{1}{2\sigma^2}\|x - t\|^2\right)h(t)^\top dt \,,
\end{aligned}
$$

with a change of variable and definition of expectation, we get the result. $\square$

**Lemma 3.** *For $L \geq 0, r \geq 0$, let $h : \mathbb{R}^d \mapsto [0, r]$ be defined as follows:*

$$h(x) = \frac{1}{2}\,sign(x_1)\min\{r, 2L|x_1|\} + \frac{r}{2} \,,$$

*where sign is the sign function with the convention $sign(0) = 0$. Then, $h$ is $L$-Lipschitz continuous.*

*Proof.* To show that $h$ is $L$-Lipschitz continuous, we need to demonstrate that for all $x, y \in \mathbb{R}^d$:

$$|h(x) - h(y)| \leq L\|x - y\|_2 \,.$$

We write $x = (x_1, \ldots, x_d)$ and $y = (y_1, \ldots, y_d)$. In the following cases, only the first coordinate is going to intervene. We consider three cases:

**Case 1:** $x_1 = 0$ and $y_1 = 0$.

In this case, $\text{sign}(x_1) = 0$, $\text{sign}(y_1) = 0$, and $h(y) = h(x) = \frac{r}{2}$ for any $x$.

Thus, $|h(x) - h(y)| = 0 \leq L\|x - y\|_2$.

**Case 2:** $x_1 \neq 0$ and $y_1 = 0$ (without loss of generality same as $x_1 = 0$ and $y_1 \neq 0$).

In this case, $h(x)$ is given by:

$$h(x) = \frac{1}{2}\text{sign}(x_1)\min\{r, 2L|x_1|\} + \frac{r}{2} \,,$$

and $h(y)$ is given by:

$$h(y) = \frac{r}{2} \, .$$

Now, let's consider the difference $|h(x) - h(y)|$:

$$|h(x) - h(y)| = \left| \frac{1}{2} \mathrm{sign}(x_1) \min\{r, 2L|x_1|\} + \frac{r}{2} - \frac{r}{2} \right| \, .$$

If $2L|x_1| \leq r$, then $\min\{r, 2L|x_1|\} = 2L|x_1|$ and the expression becomes:

$$\left| \frac{1}{2}(2Lx_1) + \frac{r}{2} - \frac{r}{2} \right| = L|x_1| \leq L\|x - y\|_2 \, .$$

If $2L|x_1| > r$, then $\min\{r, 2L|x_1|\} = r$ and the expression becomes:

$$\left| \frac{1}{2}\mathrm{sign}(x_1)r + \frac{r}{2} - \frac{r}{2} \right| = \frac{r}{2} \leq L|x_1| \leq L\|x - y\|_2 \, .$$

In both cases, $|h(x) - h(y)| \leq L\|x - y\|_2$, therefore, in the case where $x_1 \neq 0$ and $y_1 = 0$, $|h(x) - h(y)|$ is $L$-Lipschitz. Same result goes for $x_1 = 0$ and $y_1 \neq 0$.

**Case 3:** $x_1 \neq 0$ and $y_1 \neq 0$.

Without loss of generality, assume $|x_1| \geq |y_1|$. Consider

$$|h(x) - h(y)| = \frac{1}{2} \left| \mathrm{sign}(x_1) \min\{r, 2L|x_1|\} - \mathrm{sign}(y_1) \min\{r, 2L|y_1|\} \right| \, .$$

Let's consider two sub-cases:

**Sub-case 3a:** If $2L|x_1| \leq r$, then $\min\{r, 2L|x_1|\} = 2L|x_1|$.

**Sub-case 3b:** If $2L|x_1| > r$, then $\min\{r, 2L|x_1|\} = r$.

Similarly, for $|y_1|$, we have $\min\{r, 2L|y_1|\} = 2L|y_1|$ if $2L|y_1| \leq r$ and $\min\{r, 2L|y_1|\} = r$ if $2L|y_1| > r$.

In both sub-cases, we can write:

$$|h(x) - h(y)| = \frac{1}{2}|2Lx_1 - 2Ly_1| = L|x_1 - y_1| \leq L\|x - y\|_2 \, .$$

Therefore, $h$ is $L$-Lipschitz continuous. $\qquad\square$

**Theorem** (1st part). *Let* $f : \mathbb{R}^d \mapsto [0, r]$ *a Lipschitz continuous classifier and* $\tilde{f}(x) = \mathbb{E}_{\delta \sim \mathcal{N}(0, \sigma^2 I)}[f(x + \delta)]$ *the associated smoothed classifier. Then,*

$$L(\tilde{f}) \leq L(f) \, \mathrm{erf}\left( \frac{r}{2^{\frac{3}{2}} L(f)\sigma} \right) \, .$$

*Proof.* For ease of notation we note $L = L(f)$, we are interested in the following:

$$J(\sigma, L) = \sup_{h:L(h)=L} \sup_{x \in \mathbb{R}^d} \|\nabla \tilde{h}(x)\|_2 = \sup_{h:L(h)=L} \sup_{x \in \mathbb{R}^d} \sup_{v \in \mathbb{R}^d : \|v\|=1} v^\top \nabla \tilde{h}(x) \, .$$

First, we will derive an upper bound on $J(\sigma, L)$. Consider any $x \in \mathbb{R}^d$, any $h$ Lipschitz continuous s.t $h(x) \in [0, r]$, and any $v \in \mathbb{R}^d$ with $\|v\| = 1$. Any $\delta \in \mathbb{R}^d$ can be decomposed as $\delta = \delta^\perp + \tilde{\delta}$, where $\tilde{\delta} = (v^T \delta)v$ and $\delta^\perp \perp v$. Let $\delta' = \delta^\perp - \tilde{\delta}$. That is, $\delta'$ is the reflection of the vector $\delta$ with respect to the hyperplane that is normal to $v$. If $\delta \sim \mathcal{N}(0, \sigma^2 I)$, then $\delta' \sim \mathcal{N}(0, \sigma^2 I)$ because $\mathcal{N}(0, \sigma^2 I)$ is radially symmetric. Moreover, $v^T \delta' = -v^T \delta$. Hence,

$$\mathbb{E}_{\delta \sim \mathcal{N}(0, \sigma^2 I)}[v^\top \delta h(x + \delta)] = \mathbb{E}_{\delta \sim \mathcal{N}(0, \sigma^2 I)}[v^\top \delta' h(x + \delta')] = -\mathbb{E}_{\delta \sim \mathcal{N}(0, \sigma^2 I)}[v^\top \delta h(x + \delta')] \, .$$

Using the above, we have the following, using Stein's Lemma 1:

$$
\begin{aligned}
v^\top \nabla \tilde{h}(x) &= \frac{1}{\sigma^2} \mathbb{E}_{\delta \sim \mathcal{N}(0,\sigma^2 I)} [v^T \delta h(x+\delta)] \\
&= \frac{1}{2\sigma^2} \mathbb{E}_{\delta \sim \mathcal{N}(0,\sigma^2 I)} [v^T \delta (h(x+\delta) - h(x+\delta'))] \\
&\leq \frac{1}{2\sigma^2} \mathbb{E}_{\delta \sim \mathcal{N}(0,\sigma^2 I)} [|v^T \delta| \, |(h(x+\delta) - h(x+\delta'))|] \\
&\overset{(i)}{\leq} \frac{1}{2\sigma^2} \mathbb{E}_{\delta \sim \mathcal{N}(0,\sigma^2 I)} [|v^T \delta| \min\{r, 2L|v^T \delta|\}] \\
&\overset{(ii)}{=} \frac{1}{2\sigma^2} \mathbb{E}_{\delta \sim \mathcal{N}(0,\sigma^2 I)} [|\delta_1| \min\{r, 2L|\delta_1|\}] \\
&\overset{(iii)}{=} \frac{1}{2\sigma^2} \mathbb{E}_{z \sim \mathcal{N}(0,\sigma^2)} [|z| \min\{r, 2L|z|\}] \,,
\end{aligned}
$$

where $(i)$ follows from the Lipschitz assumption on $h$ and the fact that $\|\delta - \delta'\| = \|2\tilde{\delta}\| = \|2(v^T \delta)v\| = 2|v^T \delta|$ and that $|h(x+\delta) - h(x+\delta')| \leq r$, $(ii)$ follows by choosing the canonical unit vector $v = (1, 0, \ldots, 0)$ because the previous expression does not depend on the direction of $v$, and $(iii)$ follows by simply rewriting the expression in terms of $z$.

Now, we will derive a lower bound on $J(\sigma, L)$. For this, we choose a specific $\bar{h} \in [0, r]$ as $\bar{h}(x) = \frac{1}{2} \text{sign}(x_1) \min\{r, 2L|x_1|\} + r/2$, with $\text{sign}(0) = 0$. Using Lemma 3, we have that $\bar{h}$ is $L$-Lipschitz. We choose a specific $x = 0$ and specific unit vector $v_0 = (1, 0, \ldots, 0)$. For this choice, note that $\bar{h}(0) = r/2$ and $v_0^\top \delta = \delta_1$. Then,

$$
\begin{aligned}
J(\sigma, L) \geq v_0^\top \nabla \tilde{\bar{h}} &= \frac{1}{\sigma^2} \mathbb{E}_{\delta \sim \mathcal{N}(0,\sigma^2 I)} [v_0^\top \delta \bar{h}(\delta)^\top] \\
&= \frac{1}{2\sigma^2} \mathbb{E}_{\delta \sim \mathcal{N}(0,\sigma^2 I)} [|\delta_1| \min\{r, 2L|\delta_1|\}] + 0 \\
&= \frac{1}{2\sigma^2} \mathbb{E}_{z \sim \mathcal{N}(0,\sigma^2)} [|z| \min\{r, 2L|z|\}] \,.
\end{aligned}
$$

Combining the upper and lower bounds, we have the following equality:

$$
J(\sigma, L) = \frac{1}{2\sigma^2} \mathbb{E}_{z \sim \mathcal{N}(0,\sigma^2)} \left[ \min\left\{ r|z|, 2Lz^2 \right\} \right] \,. \tag{10}
$$

We will now compute the above expression exactly:

$$
\begin{aligned}
&\frac{1}{2\sigma^2} \mathbb{E}_{z \sim \mathcal{N}(0,\sigma^2)} \left[ \min\left\{ r|z|, 2Lz^2 \right\} \right] \\
&\quad = \frac{1}{2\sigma^2} \int_{-\frac{r}{2L}}^{\frac{r}{2L}} \frac{1}{\sqrt{2\pi\sigma^2}} \exp\left( \frac{-z^2}{2\sigma^2} \right) 2Lz^2 dz \\
&\qquad - \frac{1}{2\sigma^2} \int_{-\infty}^{-\frac{r}{2L}} \frac{r}{\sqrt{2\pi\sigma^2}} \exp\left( \frac{-z^2}{2\sigma^2} \right) z dz \\
&\qquad + \frac{1}{2\sigma^2} \int_{\frac{r}{2L}}^{+\infty} \frac{1}{\sqrt{2\pi\sigma^2}} \exp\left( \frac{-z^2}{2\sigma^2} \right) z dz \\
&\quad = \frac{1}{\sigma^2} \int_{-\frac{r}{2L}}^{\frac{r}{2L}} \frac{1}{\sqrt{2\pi\sigma^2}} \exp\left( \frac{-z^2}{2\sigma^2} \right) Lz^2 dz + \frac{e^{-\frac{r}{8L^2\sigma^2}}}{2^{\frac{3}{2}}\sqrt{\pi}\,|\sigma|} + \frac{e^{-\frac{r}{8L^2\sigma^2}}}{2^{\frac{3}{2}}\sqrt{\pi}\,|\sigma|} \\
&\quad = L \operatorname{erf}\left( \frac{r}{2^{\frac{3}{2}} L\sigma} \right) - \frac{e^{-\frac{r}{8L^2\sigma^2}}}{\sqrt{2}\sqrt{\pi}\,|\sigma|} + \frac{e^{-\frac{r}{8L^2\sigma^2}}}{2^{\frac{3}{2}}\sqrt{\pi}\,|\sigma|} + \frac{e^{-\frac{r}{8L^2\sigma^2}}}{2^{\frac{3}{2}}\sqrt{\pi}\,|\sigma|} \\
&\quad = L \operatorname{erf}\left( \frac{r}{2^{\frac{3}{2}} L\sigma} \right) \,.
\end{aligned}
$$

$\square$

**Remark 1.** *Jensen's inequality gives the following simple upper bound on $J(\sigma, L)$:*

$$J(\sigma, L) = \frac{1}{2\sigma^2} \mathbb{E}_{z \sim \mathcal{N}(0, \sigma^2)} \left[ \min \left\{ r|z|, 2Lz^2 \right\} \right]$$

$$\leq \min \left\{ \mathbb{E}_{z \sim \mathcal{N}(0, \sigma^2)}[r|z|/2\sigma^2], \mathbb{E}_{z \sim \mathcal{N}(0, \sigma^2)}[Lz^2/\sigma^2] \right\}$$

$$\leq \min \left\{ \frac{r}{\sqrt{2\pi\sigma^2}}, L \right\} .$$

*Hence, $J(\sigma, L)$ is no worse than the Lipschitz constant of the original classifier $f$, or its Gaussian smoothed counterpart without the Lipschitz assumption, the latter bound is twice smaller than the previous original derivation in (Salman et al., 2019, Appendix A).*

**Theorem** (2nd part). *Let $f : \mathbb{R}^d \mapsto \Delta_r^{c-1}$ a Lipschitz continuous classifier and $\tilde{h}(x) = \mathbb{E}_{\delta \sim \mathcal{N}(0, \sigma^2 I)}[f(x + \delta)]$ the associated smoothed classifier. Then,*

$$L(\tilde{f}) \leq L(f) \operatorname{erf}\left( \frac{r}{2^{\frac{3}{2}} L(f)\sigma} \right) .$$

*Proof.* We also note here $L = L(f)$ and use the same notation as in the previous proof. Here $h : \mathbb{R}^d \mapsto \Delta^{c-1}$, we have to consider a bound on the maximum singular value of the Jacobian,

$$J(\sigma, L) = \sup_{h:L(h)=L} \sup_{x \in \mathbb{R}^d} \sup_{\substack{v \in \mathbb{R}^d : \|v\|=1 \\ u \in \mathbb{R}^c : \|u\|=1}} v^\top \frac{\partial}{\partial x} \tilde{h}(x) u .$$

As in the previous proof, we will derive an upper bound on $J(\sigma, L)$. Consider any $x \in \mathbb{R}^d$, any $h$ $L$-Lipschitz continuous s.t $h(x) \in \Delta_r^{c-1}$, and any $v \in \mathbb{R}^d$ with $\|v\| = 1$, $u \in \mathbb{R}^c$ with $\|u\| = 1$.

Any $\delta \in \mathbb{R}^d$ can be decomposed as $\delta = \delta^\perp + \tilde{\delta}$, where $\tilde{\delta} = (v^T \delta)v$ and $\delta^\perp \perp v$. Let $\delta' = \delta^\perp - \tilde{\delta}$. That is, $\delta'$ is the reflection of the vector $\delta$ with respect to the hyperplane that is normal to $v$. If $\delta \sim \mathcal{N}(0, \sigma^2 I)$, then $\delta' \sim \mathcal{N}(0, \sigma^2 I)$ because $\mathcal{N}(0, \sigma^2 I)$ is radially symmetric. Moreover, $v^T \delta' = -v^T \delta$. Hence,

$$\mathbb{E}_{\delta \sim \mathcal{N}(0, \sigma^2 I)}[v^\top \delta h(x + \delta)^\top u] = \mathbb{E}_{\delta \sim \mathcal{N}(0, \sigma^2 I)}[v^\top \delta' h(x + \delta')^\top u] = -\mathbb{E}_{\delta \sim \mathcal{N}(0, \sigma^2 I)}[v^\top \delta h(x + \delta')^\top u] .$$

Using the above, we have the following, using extended Stein's Lemma 2:

$$v^\top \frac{\partial}{\partial x} \tilde{h}(x)^\top v = \frac{1}{\sigma^2} \mathbb{E}_{\delta \sim \mathcal{N}(0, \sigma^2 I)}[v^T \delta h(x + \delta)^\top u]$$

$$= \frac{1}{2\sigma^2} \mathbb{E}_{\delta \sim \mathcal{N}(0, \sigma^2 I)}[v^T \delta (h(x + \delta) - h(x + \delta')^\top u)]$$

$$\leq \frac{1}{2\sigma^2} \mathbb{E}_{\delta \sim \mathcal{N}(0, \sigma^2 I)}[|v^T \delta| \, |(h(x + \delta) - h(x + \delta'))^\top u|]$$

$$\stackrel{(i)}{\leq} \frac{1}{2\sigma^2} \mathbb{E}_{\delta \sim \mathcal{N}(0, \sigma^2 I)}[|v^T \delta| \min\{r\sqrt{2}, 2L|v^T \delta|\}]$$

$$\stackrel{(ii)}{=} \frac{1}{2\sigma^2} \mathbb{E}_{\delta \sim \mathcal{N}(0, \sigma^2 I)}[|\delta_1| \min\{r\sqrt{2}, 2L|\delta_1|\}]$$

$$\stackrel{(iii)}{=} \frac{1}{2\sigma^2} \mathbb{E}_{z \sim \mathcal{N}(0, \sigma^2)}[|z| \min\{r\sqrt{2}, 2L|z|\}] ,$$

where $(i)$ follows from the Lipschitz assumption on $h$ and the fact that $\|\delta - \delta'\| = \|2\tilde{\delta}\| = \|2(v^T \delta)v\| = 2|v^T \delta|$ and that $|(h(x + \delta) - h(x + \delta'))^\top u| = \|(h(x + \delta) - h(x + \delta'))^\top u\| \leq \|h(x + \delta) - h(x + \delta')\| \|u\| \leq r\sqrt{2}$, $(ii)$ follows by choosing the canonical unit vector $v = (1, 0, \ldots, 0)$ because the previous expression does not depend on the direction of $v$, and $(iii)$ follows by simply rewriting the expression in terms of $z$.

Now, we will derive a lower bound on $J(\sigma, L)$. For this, we choose a specific $\bar{h} \in [0, r]^c$ as $\bar{h}_1(x) = \operatorname{sign}(x_1) \min\{\frac{r\sqrt{2}}{2}, L|x_1|\} + \frac{r\sqrt{2}}{2}$, with $\operatorname{sign}(0) = 0$ and $\bar{h}_i(x) = 0$ for $i \in \{2, \ldots c\}$. Using Lemma 3 $\bar{h}_1$ is $L$-Lipschitz and $h_i$ are 0-Lipschitz for $i \in \{2, \ldots c\}$, thus $\bar{h}$ is $L$-Lipschitz.

a specific $x = 0$ and specific unit vectors $\bar{u} = (1, 0, \ldots, 0)$, $\bar{v} = (1, 0, \ldots, 0)$. For this choice, note that $\bar{h}_1(0) = r\sqrt{2}/2$ and $\bar{v}^\top \delta = \delta_1$. Then,

$$
\begin{aligned}
J(\sigma, L) \geq \bar{v}^\top \frac{\partial}{\partial x} \tilde{\bar{h}}(0)\bar{u} &= \frac{1}{\sigma^2} \mathbb{E}_{\delta \sim \mathcal{N}(0, \sigma^2 I)}[\bar{v}^\top \delta \bar{h}(\delta)^\top \bar{u}] \\
&= \frac{1}{2\sigma^2} \mathbb{E}_{\delta \sim \mathcal{N}(0, \sigma^2 I)}[|\delta_1| \min\{r\sqrt{2}, 2L|\delta_1|\}] + 0 \\
&= \frac{1}{2\sigma^2} \mathbb{E}_{z \sim \mathcal{N}(0, \sigma^2)}[|z| \min\{r\sqrt{2}, 2L|z|\}] .
\end{aligned}
$$

Combining the upper and lower bounds, we have the following equality:

$$
J(\sigma, L) = \frac{1}{2\sigma^2} \mathbb{E}_{z \sim \mathcal{N}(0, \sigma^2)} \left[ \min\left\{ r\sqrt{2}|z|, 2L(h)z^2 \right\} \right] . \tag{11}
$$

We will now compute the above expression exactly:

$$
\begin{aligned}
\frac{1}{2\sigma^2} \mathbb{E}_{z \sim \mathcal{N}(0, \sigma^2)} &\left[ \min\left\{ \sqrt{2}|z|, 2Lz^2 \right\} \right] \\
&= \frac{1}{2\sigma^2} \int_{-\frac{r}{\sqrt{2}L}}^{\frac{r}{\sqrt{2}L}} \frac{1}{\sqrt{2\pi\sigma^2}} \exp\left( \frac{-z^2}{2\sigma^2} \right) 2Lz^2 dz \\
&\quad - \frac{1}{2\sigma^2} \int_{-\infty}^{-\frac{r}{\sqrt{2}L}} \frac{1}{\sqrt{2\pi\sigma^2}} \exp\left( \frac{-z^2}{2\sigma^2} \right) z dz \\
&\quad + \frac{1}{2\sigma^2} \int_{\frac{r}{\sqrt{2}L}}^{+\infty} \frac{1}{\sqrt{2\pi\sigma^2}} \exp\left( \frac{-z^2}{2\sigma^2} \right) z dz \\
&= \frac{1}{\sigma^2} \int_{-\frac{r}{\sqrt{2}L}}^{\frac{r}{\sqrt{2}L}} \frac{1}{\sqrt{2\pi\sigma^2}} \exp\left( \frac{-z^2}{2\sigma^2} \right) Lz^2 dz + \frac{e^{-\frac{r}{4L^2\sigma^2}}}{2^{\frac{3}{2}}\sqrt{\pi}|\sigma|} + \frac{e^{-\frac{r}{4L^2\sigma^2}}}{2^{\frac{3}{2}}\sqrt{\pi}|\sigma|} \\
&= L \operatorname{erf}\left( \frac{r}{2L\sigma} \right) - \frac{e^{-\frac{r}{4L^2\sigma^2}}}{\sqrt{2}\sqrt{\pi}|\sigma|} + \frac{e^{-\frac{r}{4L^2\sigma^2}}}{2^{\frac{3}{2}}\sqrt{\pi}|\sigma|} + \frac{e^{-\frac{r}{4L^2\sigma^2}}}{2^{\frac{3}{2}}\sqrt{\pi}|\sigma|} \\
&= L \operatorname{erf}\left( \frac{r}{2L\sigma} \right) .
\end{aligned}
$$

$\square$

**Remark 2.** *For $h : \mathbb{R}^d \mapsto \Delta^{c-1}$ a Lipschitz classifier, Jensen's inequality gives the following simple upper bound on $J(\sigma, L)$:*

$$
\begin{aligned}
J(\sigma, L) &= \frac{1}{2\sigma^2} \mathbb{E}_{z \sim \mathcal{N}(0, \sigma^2)} \left[ \min\left\{ r\sqrt{2}|z|, 2Lz^2 \right\} \right] \\
&\leq \min\left\{ \mathbb{E}_{z \sim \mathcal{N}(0, \sigma^2)}[r\sqrt{2}|z|/2\sigma^2], \mathbb{E}_{z \sim \mathcal{N}(0, \sigma^2)}[Lz^2/\sigma^2] \right\} \\
&\leq \min\left\{ \frac{r}{\sqrt{\pi\sigma^2}}, L \right\} .
\end{aligned}
$$

*Hence, $J(\sigma, L)$ is no worse than the Lipschitz constant of the original classifier $h$, or its Gaussian smoothed counterpart without the Lipschitz assumption, the latter bound is $\sqrt{2}$ times bigger than the bi-class case with $L(\tilde{h}_k)$.*

## D.2 Proof of Proposition 3

**Proposition.** *The optimal value $\sigma^*$ that maximizes the gap between the bounds of Eq. (5) is:*

$$
\sigma^* = \frac{r}{L(s_k^r \circ f)\sqrt{2\pi}} \quad \text{giving} \quad L(\tilde{f}_k) \leq \operatorname{erf}(\sqrt{\pi}/2) \, L(s_k^r \circ f) \lesssim 0.79 \, L(s_k^r \circ f) .
$$

*Similarly, for Eq. (6):*

$$
\sigma^* = \frac{r}{L(s^r \circ f)\sqrt{\pi}} \quad \text{giving} \quad L(\tilde{f}) \leq \operatorname{erf}(\sqrt{\pi}/2) \, L(s^r \circ f) \lesssim 0.79 \, L(s^r \circ f) .
$$

*Proof.* For $\gamma \geq 0$, we seek $\sigma^*$ that maximizes the gap between the bounds of Eq. (5) with respect to $\sigma$:

$$\sigma^* = \arg\max_{\sigma > 0} \left\{ \min \left\{ \gamma, \frac{r}{\sqrt{2\pi\sigma^2}} \right\} - \gamma \operatorname{erf}\left( \frac{r}{2^{3/2}\gamma\sigma} \right) \right\} .$$

To find the value of $\sigma^*$ that maximizes the given function, we'll determine the critical points. Let

$$g(\sigma) = \min \left\{ \gamma, \frac{r}{\sqrt{2\pi\sigma^2}} \right\} - \gamma \operatorname{erf}\left( \frac{r}{2^{3/2}\gamma\sigma} \right) .$$

Let's start by setting the two functions inside the $\min$ function equal to each other and solving for $\sigma$:

$$\gamma = \frac{r}{\sqrt{2\pi\sigma^2}}$$

$$\Rightarrow \sigma^2 = \frac{r}{\gamma^2 2\pi}$$

$$\Rightarrow \sigma = \frac{r}{\gamma\sqrt{2\pi}} .$$

This is the point of intersection, hence the value of $\sigma$ where the two functions inside the $\min$ change dominance. For that value, $g(\frac{r}{\gamma\sqrt{2\pi}}) = \gamma(1 - \operatorname{erf}(\frac{\sqrt{\pi}}{2}))$ .

Now, for $\sigma < \frac{r}{\gamma\sqrt{2\pi}}$, $g(\sigma) = \gamma - \gamma \operatorname{erf}\left( \frac{r}{2^{3/2}\gamma\sigma} \right)$ .

Let's differentiate $g(\sigma)$ in the this first region:

$$g'(\sigma) = 0 - \frac{d}{d\sigma}\left[ \gamma \operatorname{erf}\left( \frac{r}{2^{3/2}\gamma\sigma} \right) \right]$$

$$= \gamma \frac{r}{2^{3/2}\gamma} \frac{2}{\sqrt{\pi}} \exp\left( -\left( \frac{r}{2^{3/2}\gamma\sigma} \right)^2 \right)$$

$$= \frac{r}{\sqrt{2\pi}} \exp\left( -\left( \frac{r}{2^{3/2}\gamma\sigma} \right)^2 \right) .$$

The supremum is obtained for $\sigma \to 0$, and the associated limit value for $g(\sigma)$ is 0.

For the second region, $\sigma > \frac{r}{\gamma\sqrt{2\pi}}$, $g(\sigma) = \frac{r}{\sqrt{2\pi\sigma^2}} - \gamma \operatorname{erf}\left( \frac{r}{2^{3/2}\gamma\sigma} \right)$. Supremum is obtained for $\sigma \to \frac{r}{\gamma\sqrt{2\pi}}$ as $g$ is a decreasing function of $\sigma$. The associated limit value for $g(\sigma)$ is $\gamma(1 - \operatorname{erf}(\frac{\sqrt{\pi}}{2}))$.

Finally, taking $\gamma = L(s_k^r \circ f)$, $\sigma^* = \frac{r}{L(s_k^r \circ f)\sqrt{2\pi}}$ gives maximum value for $g$ on all domain.

We get a similar result $\sigma^* = \frac{r}{L(s^r \circ f)\sqrt{\pi}}$ for $L(s^r \circ f)$. $\qquad\square$

### D.3 PROOF OF THEOREM 3

In the previous section, we derived a new radius for the smoothed classifier $\tilde{h}$ but usually RS approaches use the Lipschitz constant of the function $\Phi^{-1} \circ \tilde{h}$ and its associated certified radius.

Here we suppose that $\tilde{h} : \mathbb{R}^d \mapsto \Delta_r^{c-1}$, how does it changes the Lipschitz constant of $\Phi^{-1} \circ \tilde{h}$ ?

**Lemma 4.** *Let* $\tilde{h} : \mathbb{R}^d \mapsto \Delta_r^{c-1}$ *be the smoothed classifier and* $\Phi^{-1}$ *the gaussian quantile function. For an input* $x \in \mathcal{X}$, *the* $\ell_2$-*norm of the gradient of* $\Phi^{-1} \circ \tilde{h}_k$ *is bounded by:*

$$\|\nabla \Phi^{-1} \circ \tilde{h}_k(x)\|_2 \leq \frac{r}{\sigma} \exp\left( -\frac{1}{2}\left( \Phi^{-1}(\tilde{h}_k(x)/r)^2 - \Phi^{-1}(\tilde{h}_k(x))^2 \right) \right) .$$

*Proof.* In the same manner as the proof of Salman et al. (2019), let us assume that $\sigma = 1$.

$$\|\nabla\Phi^{-1}\circ\tilde{h}_k(x)\|_2 = \sup_{v\in\mathbb{R}^d:\|v\|=1} v^\top\nabla\Phi^{-1}(\tilde{h}_k(x))$$

$$= \sup_{v\in\mathbb{R}^d:\|v\|=1} \frac{v^\top\nabla\tilde{h}_k(x)}{\Phi'(\Phi^{-1}(\tilde{h}_k(x)))}$$

Using that expression, we can derive an upper bound on the Lipschitz constant of $\Phi^{-1}\circ\tilde{h}_k$.

The denominator $\Phi'(\Phi^{-1}(\tilde{h}_k(x))) = \frac{1}{\sqrt{2\pi}}\exp\left(-\frac{1}{2}\left(\Phi^{-1}(\tilde{h}_k(x))^2\right)\right)$.

By Stein's Lemma, we can express the numerator as

$$v^\top\nabla\tilde{h}_k(x) = \mathbb{E}_{\delta\sim\mathcal{N}(0,I)}[v^\top\delta h_k(x+\delta)].$$

We need to bound this quantity with the constraint that $\mathbb{E}_{\delta\sim\mathcal{N}(0,I)}[h_k(x+\delta)] = p$ and $h_k(x)\in[0,r]$, it sums to following problem, with $h_k(x+z) = g(z)$:

$$\sup_{\substack{g\\ v\in\mathbb{R}^d:\|v\|=1}} \mathbb{E}_{\delta\sim\mathcal{N}(0,I)}\left[v^\top\delta g(x)\right] \tag{12}$$

$$\text{s.t}\quad g(x)\in[0,r]\ \text{ and }\ \mathbb{E}_{\delta\sim\mathcal{N}(0,I)}\left[g(x)\right] = p.$$

We can solve it for $g' = g/r$:

$$\sup_{\substack{g'\\ v\in\mathbb{R}^d:\|v\|=1}} r\mathbb{E}_{\delta\sim\mathcal{N}(0,I)}\left[v^\top\delta g'(x)\right]$$

$$\text{s.t}\quad g'(x)\in[0,1]\ \text{ and }\ \mathbb{E}_{\delta\sim\mathcal{N}(0,I)}\left[g'(x)\right] = p/r.$$

We recognize problem solved in (Salman et al., 2019, Appendix A, Lemma 2), which has for solution $g'^*(z) = \mathbb{1}_{v^\top z>\Phi^{-1}(p/r)}$. Thus the problem (12) has for solution $g^*(z) = r\mathbb{1}_{v^\top z>\Phi^{-1}(p/r)}$.

Plugging $g^*$ in the numerator we obtain

$$\mathbb{E}_{\delta\sim\mathcal{N}(0,I)}[v^\top\delta g^*(\delta)] = r\,\mathbb{E}_{Z\sim\mathcal{N}(0,1)}[Z\,\mathbb{1}_{Z>\Phi^{-1}(p/r)}]$$

$$= \frac{r}{\sqrt{2\pi}}\int_{-\Phi^{-1}(p/r)}^{\infty} t\exp{-t^2/2}dt$$

$$= \frac{r}{\sqrt{2\pi}}\exp\left(-\frac{1}{2}\Phi^{-1}(\tilde{h}_k(x)/r)^2\right).$$

Finally,

$$\|\nabla\Phi^{-1}\circ\tilde{h}_k(x)\|_2 \le r\exp\left(-\frac{1}{2}(\Phi^{-1}(\tilde{h}_k(x)/r)^2 - \Phi^{-1}(\tilde{h}_k(x))^2)\right).$$

To obtain the result for any $\sigma$, we can apply the result to $\tilde{h}_k(x/\sigma)$ and this implies that

$$\|\nabla\Phi^{-1}\circ\tilde{h}_k(x)\|_2 \le \frac{r}{\sigma}\exp\left(-\frac{1}{2}(\Phi^{-1}(\tilde{h}_k(x)/r)^2 - \Phi^{-1}(\tilde{h}_k(x))^2)\right).$$

$\square$

Using previous Lemma 4 we derive the following Theorem,

**Theorem.** *Let* $\tilde{h}:\mathbb{R}^d\mapsto\Delta_r^{c-1}$ *be the smoothed classifier and* $\Phi^{-1}$ *the gaussian quantile function. For an input* $x\in\mathcal{X}$, *and* $\mathcal{B} = B_2\left(\tilde{h}(x),\epsilon L(\tilde{h})\right)$ *the* $\ell_2$*-norm of the local Lipschitz constant of* $\Phi^{-1}\circ\tilde{h}_k$ *is bounded by:*

$$L\left(\nabla\Phi^{-1}\circ\tilde{h}_k,\mathcal{B}\right) \le \frac{r}{\sigma}\max_{p\in\mathcal{B}}\left\{\exp\left(-\frac{1}{2}\left(\Phi^{-1}(p/r)^2 - \Phi^{-1}(p)^2\right)\right)\right\}.$$

# E  LVM-RS Algorithm

---

**Algorithm 2** LVM-RS ($f, \sigma, x, n_0, n$)

---

1: scores_$n_0 \leftarrow$ SampleScores($f, x, n_0, \sigma$)  // validation set, dimension $n_0 \times c$
2: scores_$n \leftarrow$ SampleScores($f, x, n, \sigma$)  // certification set, dimension $n \times c$
3: **for** temperature $t \in [t_{\text{lower}}, t_{\text{upper}}]$
4:   **for** simplex map $s \in \mathcal{S}$
5:     $\bar{p}_{s^t} = \frac{1}{n_0} \sum_{i=1}^{n_0} s^t(\text{scores}\_n_0[i,:]) - \text{shift}(S_{n0}(s^t(\text{scores}\_n_0)), \alpha, n_0)$  // dimension $c$
6: $(s_*, t_*) = \arg\max_{s,t} R_2(\bar{p}_{s^t})$
7: $\bar{p}^* = \frac{1}{n_0} \sum_{i=1}^{n_0} s_*^{t_*}(\text{scores}\_n[i,:]) - \text{shift}(S_n(s_*^{t_*}(\text{scores}\_n)), \alpha, n)$  // dimension $c$
8: **return** prediction $\arg\max_k \bar{p}_k^*$ and certified radius $R_2(\bar{p}^*)$

---

We recall that RS produces the smoothed classifier $\tilde{F}$ starting from sub classifier $f$, and for all $x \in \mathcal{X}$, it outputs a certified radius $R = R(\tilde{F}, x)$ and a prediction $\tilde{F}(x) = \hat{y}$, it is guaranteed that for all $x' \in B_2(x, R), \tilde{F}(x') = \hat{y}$.

The difference with the LVM-RS procedure is that the choice of produced classifier $\tilde{F}$ depends on input $x$. Starting from a sub-classifier $f$, we generate an ensemble of smoothed classifiers $\{\tilde{F}_s\}_s$. For an input $x \in \mathcal{X}$, the LVM-RS procedure selects a classifier $\tilde{F} \in \{\tilde{F}_s\}_s$ that maximizes the margin-variance trade-off. We output for $x$ and $\tilde{F}$ a certified radius $R = R(\tilde{F}, x)$ and a prediction $\tilde{F}(x) = \hat{y}$, it is guaranteed that for all $x' \in B_2(x, R), \tilde{F}(x') = \hat{y}$.

## F ABLATION STUDY

This ablation study provides two comparisons:

- A comparison between corrected certified radii produced by Hoeddfing's and Bernstein's inequalities in Fig. 2. The Clopper-Pearson is not included as it is only applicable to binomial values.
- A comparison between corrected certified radii produced by different simplex maps and temperatures in Fig.3.

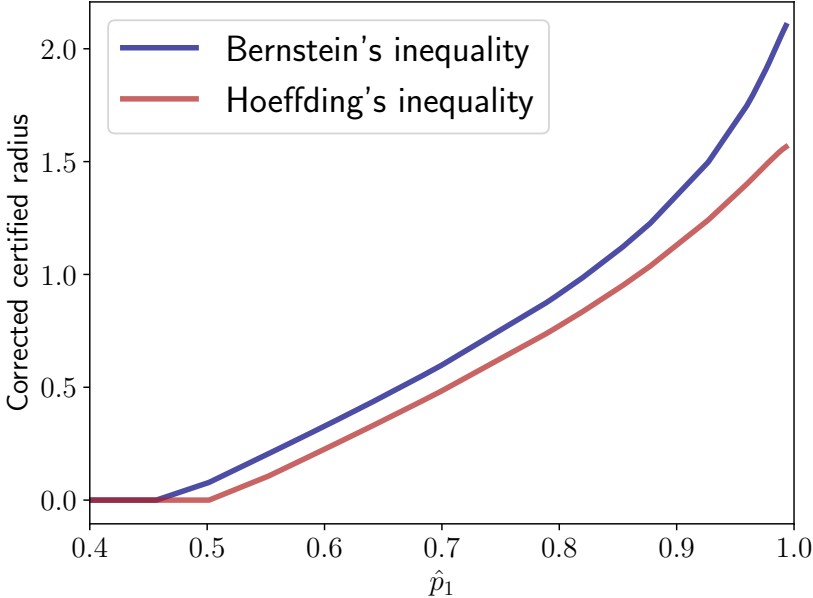

Figure 2: Comparison between corrected certified radii $R_2(\bar{p})$ produced by Bernstein's and Hoeffding's inequalities, for a random subset of 1000 images of ImageNet dataset using RS with a smoothing noise $\sigma = 1.0$. We use the ViT-denoiser baseline from Carlini et al. (2023).

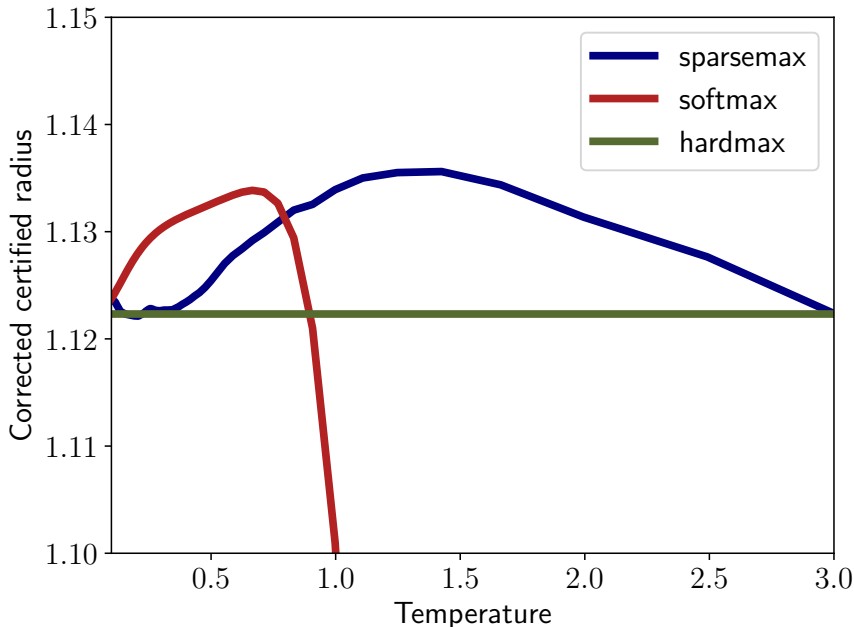

Figure 3: Comparison of the effect on corrected certified radii $R_2(\bar{p})$ of the choice of the simplex map $s$ and associated temperature $t$. Simplex maps considered are $s \in \{\mathrm{sparsemax}, \mathrm{softmax}, \mathrm{hardmax}\}$. The *base subclassifier* is the one from Carlini et al. (2023) and the corrected certified radii were generated with one image from ImageNet with smoothing variance $\sigma = 1.0$. Radii are risk corrected with Empirical Bernstein inequality for a risk $\alpha = 1e{-}3$ and $n = 10^4$. We see that by varying the temperature $t$, softmax and sparsemax can find a better solution than hardmax to the variance-margin trade-off.

## G FIGURES AND TABLES FOR EXPERIMENT 4.2

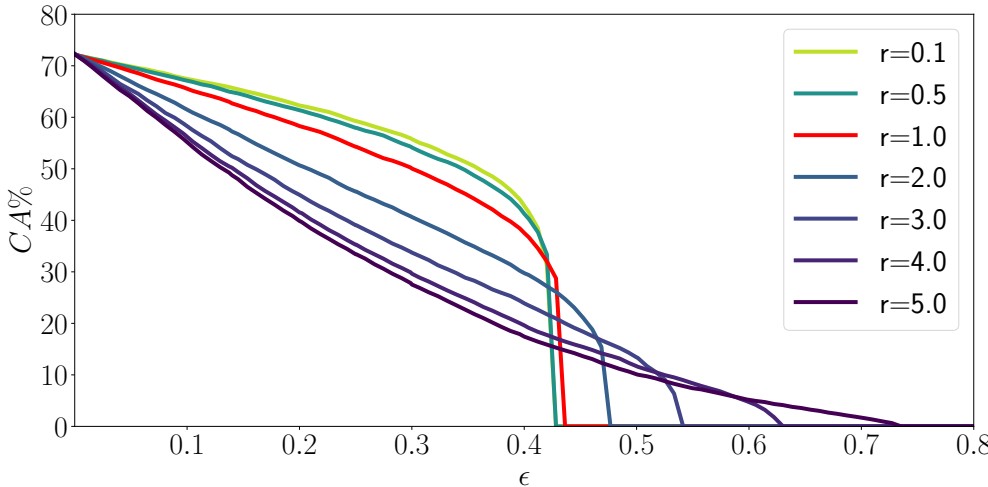

Figure 4: Certified accuracies ($CA$ in %) with $R_1$ in function of levels of perturbation $r$ on CIFAR-10, for different simplex mass $r$. Number of samples is $n = 10^4$ and risk $\alpha = 1e$-3. The case $r = 1.0$ corresponds to the regular RS setting.

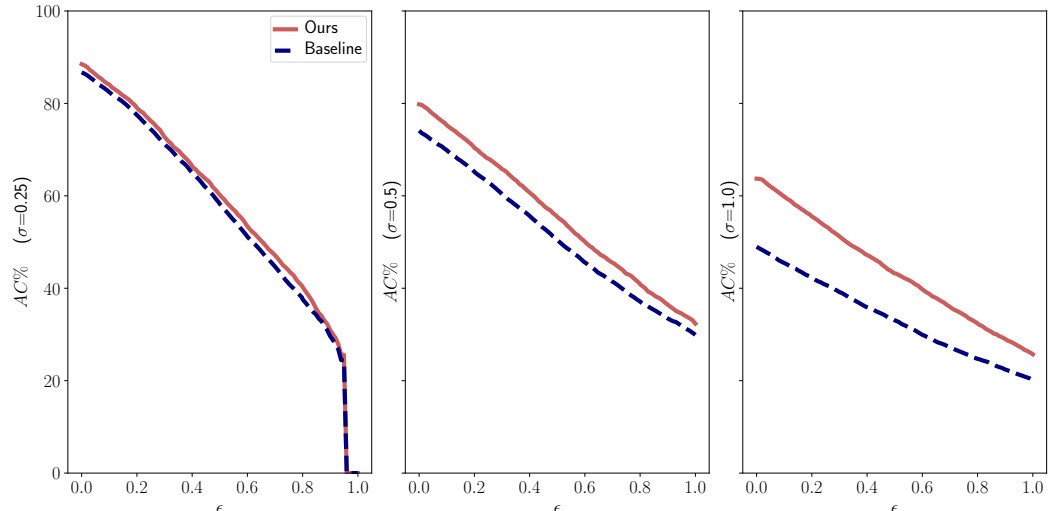

Figure 5: Certified accuracies ($CA$ in %) in function of level of perturbations $\epsilon$ on CIFAR-10, for different noise levels $\sigma = \{0.25, 0.5, 1\}$. Number of samples is $n = 10^5$ and risk $\alpha = 1e\text{-}3$. Our method is compared to the baseline chosen as in Carlini et al. (2023).

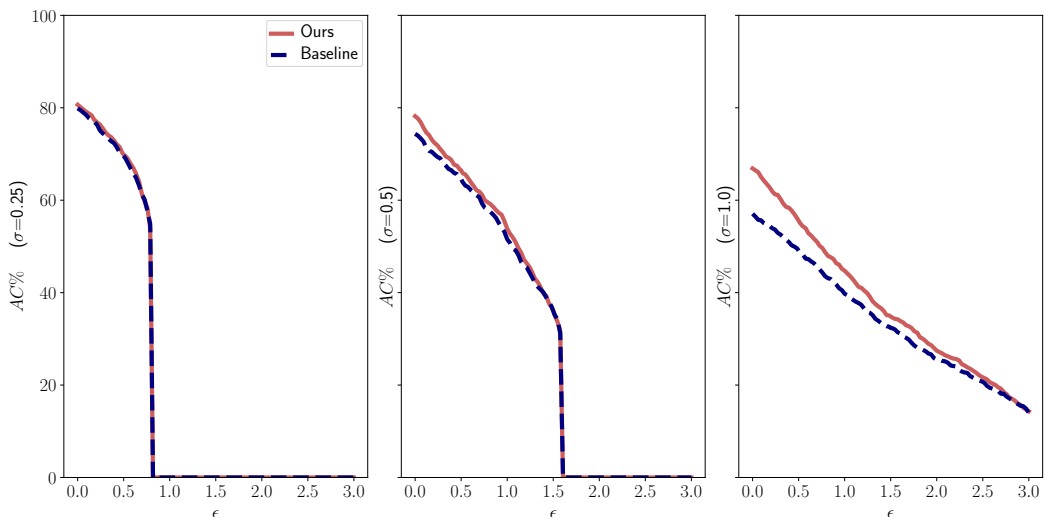

Figure 6: Certified accuracies ($CA$ in %) in function of level of perturbations $\epsilon$ on ImageNet, for different noise levels $\sigma = \{0.25, 0.5, 1\}$. Number of samples is $n = 10^4$ and risk $\alpha = 1e\text{-}3$. Our method is compared to the baseline chosen as in Carlini et al. (2023).

Table 5: Certified accuracies comparison for different perturbation $\epsilon$ values, for $n = 10^4$ samples and $\alpha = 1e\text{-}3$. On ImageNet dataset. Here the baseline is a ResNet-150 from Salman et al. (2019).

| Methods | Certified accuracy ($\varepsilon$) | | | | | | | |
|---|---|---|---|---|---|---|---|---|
| | 0.14 | 0.2 | 0.3 | 0.4 | 0.5 | 0.6 | 0.7 | 0.8 |
| Salman et al. (2019) | 74.49 | 73.08 | 69.84 | 66.41 | 62.42 | 57.75 | 51.24 | 0.0 |
| **LVM-RS (ours)** | **76.77** | **74.99** | **71.26** | **67.55** | **63.43** | **58.59** | **51.39** | 0.0 |

Table 6: Certified accuracy for $\sigma = 0.25$ on CIFAR-10, for risk $\alpha = 1e-3$ and $n = 10^5$ samples.

| Methods | Certified accuracy ($\varepsilon$) | | | | | | | | | | |
|---|---|---|---|---|---|---|---|---|---|---|---|
| | 0.0 | 0.14 | 0.2 | 0.25 | 0.3 | 0.4 | 0.5 | 0.6 | 0.75 | 0.8 | 1.0 |
| Carlini et al. (2023) | 86.72 | 80.73 | 77.47 | 74.41 | 71.15 | 65.01 | 58.25 | 51.15 | 40.96 | 37.6 | 0.0 |
| **LVM-RS (ours)** | **88.49** | **82.15** | **79.06** | **76.21** | **72.73** | **66.41** | **60.22** | **53.41** | **43.76** | **40.27** | 0.0 |

Table 7: Certified accuracy for $\sigma = 0.5$ on CIFAR-10, for risk $\alpha = 1e-3$ and $n = 10^5$ samples.

| Methods | Certified accuracy ($\varepsilon$) | | | | | | | | | | |
|---|---|---|---|---|---|---|---|---|---|---|---|
| | 0.0 | 0.14 | 0.2 | 0.25 | 0.3 | 0.4 | 0.5 | 0.6 | 0.75 | 0.8 | 1.0 |
| Carlini et al. (2023) | 74.11 | 67.99 | 65.22 | 62.89 | 60.38 | 55.67 | 50.43 | 45.59 | 39.26 | 37.11 | 29.91 |
| **LVM-RS (ours)** | **79.79** | **73.45** | **70.41** | **68.04** | **65.8** | **60.71** | **55.48** | **50.07** | **43.13** | **40.83** | **32.35** |

Table 8: Certified accuracy for $\sigma = 1$ on CIFAR-10, for risk $\alpha = 1e-3$ and $n = 10^5$ samples.

| Methods | Certified accuracy ($\varepsilon$) | | | | | | | | | | |
|---|---|---|---|---|---|---|---|---|---|---|---|
| | 0.0 | 0.14 | 0.2 | 0.25 | 0.3 | 0.4 | 0.5 | 0.6 | 0.75 | 0.8 | 1.0 |
| Carlini et al. (2023) | 48.97 | 44.24 | 42.26 | 40.76 | 39.15 | 35.91 | 33.08 | 29.92 | 25.97 | 24.72 | 20.09 |
| **LVM-RS (ours)** | **63.72** | **57.99** | **55.54** | **53.4** | **51.23** | **47.19** | **43.19** | **39.76** | **34.27** | **32.35** | **25.71** |

Table 9: Certified accuracy for $\sigma = 0.25$ on ImageNet, for risk $\alpha = 1e-3$ and $n = 10^4$ samples.

| Methods | Certified accuracy ($\varepsilon$) | | | | | |
|---|---|---|---|---|---|---|
| | 0.0 | 0.5 | 1.0 | 1.5 | 2 | 3 |
| Carlini et al. (2023) | 79.88 | 69.57 | 0.0 | 0.0 | 0.0 | 0.0 |
| **LVM-RS (ours)** | **80.66** | **69.84** | 0.0 | 0.0 | 0.0 | 0.0 |

Table 10: Certified accuracy for $\sigma = 0.5$ on ImageNet, for risk $\alpha = 1e-3$ and $n = 10^4$ samples.

| Methods | Certified accuracy ($\varepsilon$) | | | | | |
|---|---|---|---|---|---|---|
| | 0.0 | 0.5 | 1.0 | 1.5 | 2 | 3 |
| Carlini et al. (2023) | 74.37 | 64.56 | 51.55 | **36.04** | 0.0 | 0.0 |
| **LVM-RS (ours)** | **78.18** | **66.47** | **53.85** | **36.04** | 0.0 | 0.0 |

Table 11: Certified accuracy for $\sigma = 1$ on ImageNet, for risk $\alpha = 1e-3$ and $n = 10^4$ samples.

| Methods | Certified accuracy ($\varepsilon$) | | | | | |
|---|---|---|---|---|---|---|
| | 0.0 | 0.5 | 1.0 | 1.5 | 2 | 3 |
| Carlini et al. (2023) | 57.06 | 49.05 | 39.74 | 32.33 | 25.53 | 14.01 |
| **LVM-RS (ours)** | **66.87** | **55.56** | **44.74** | **34.83** | **27.43** | **14.31** |

