# OpenReview forum: "The Lipschitz-Variance-Margin Tradeoff for Enhanced Randomized Smoothing"
_ICLR.cc/2024/Conference — ICLR 2024 poster_

### Official Review · Reviewer_W4Xb · 2023-10-31

**Soundness:** 3 good
**Presentation:** 3 good
**Contribution:** 3 good
**Rating:** 6
**Confidence:** 4

**Summary:**

This work analyzes the certified radius that can be provided to off-the-shelf classifier that are also Lipschitz continuous. The authors use previous results (namely, that a certified radius can be derived deterministically for Lipschitz predictors, and that randomized smoothing provides probabilistic certified radius as well as Lipschitz smoothed predictors) to show that there exist an optimal way to determine the strength of randomized smoothing that should be applied to a Lipschitz function. In addition, the authors make the observation of changing  the mapping function to the probability simplex as well as better finite-sample concentration inequalities to derived slightly improved guarantees on popular benchmarks (CIFAR10 and ImageNet).

**Strengths:**

- While it has been known that randomized smoothing provides a Lipchitz continuous randomized classifier, with constant inversely proportional to the smoothing strength, it has been unclear what one can gain by incorporating the (natural) assumption that base (or sub) classifier is also Lipschitz. In my opinion, this is the main contribution of this paper, which is nice.

**Weaknesses:**

- The paper is at times verbose, and at times unclear. Some comments are exaggerated or incorrect (See below).
- Nomenclature is often incorrect (see below).
- In my view, the contributions of employing a better concentration inequality to provide tighter tail guarantees for finite samples is small, as is the fact of changing the function that maps the sub-classifier predictions to the probability simplex.

**Questions:**

**Major Points**

* Throughout the paper, the authors refer to softmax (and other functions) as "projections" onto the probability simplex. The softmax is not a projection onto the simplex. In fact, this reviewer believes that the softmax is not a projection onto any set (is it?). Throughout the manuscript, the word "projection" is used in lieu of the term "function" or "map".

* Randomized smoothing can provide a certified radius at varying levels, depending on the strength of the added noise. It is clear to me that leveraging the fact that $f$ is Lipschitz continuous allows the authors to find an optimal randomized smoothing strength, as indicated in Prop. 4. Such a smoothing strength results in a certified radius. What is unclear to me is, what can be done if one requires certification at a different radius?

* page 3: The definition of $L(f)$ that the authors use require $f$ to be differentiable. This assumption is not necessary (as best as I can tell) for their results to hold (and indeed, not applicable to ReLU networks). So, just define $L(f)$ by its standard definition (without assuming differentiability).

* The authors comment on some related works pertaining the analysis of Lipschitz-continuity for devising certified radii. There exist extensions of these ideas that refine the analysis to requiring only local Lipschitzness, as in [A], which lead to improvements on the certified radius. Can the authors comment on relations to that work, and on whether their results could apply in a local way as in [A] ?

* page 4: the authors mention that "these bounds are relevant for the composite classifier defined as $\Phi^{-1} \circ f$. However, I believe they mean the classifier $\tilde{F}_s$, no?

* The authors argue in Section 3.3 that different functions s(z) have different impacts in terms of the variance of s(Z) vs the variance in Z. This is quite obvious, and, respectfully, I find the Example 1 not useful/trivial. ii) It is unclear, however, how such a map should be chosen to be optimal. It seems to me (from their experimental section) that they simply compute different options based on different alternatives for s(z) and choose the one that results in larger empirical margin. Is this correct? Shouldn't one be able to provide a "best" choice for s(z)?

* After Prop. 3, the authors mention that "This refinement on the bound was possible by supposing Lipschitz continuity on the base classifier $f$. Note that its Lipschitz constant can be arbitrarily high, so this assumption is quite light". What do the authors mean by "light" here?


References:

(A) Muthukumar et al, "Adversarial Robustness of Sparse Local Lipschitz Predictors, SIAM Journal on Mathematics of Data Science, 5:4, 2023



**Other points**:
* page 1: the authors write that there's a "circular dependency: the regularity of the smoothed classifier depends on the Lipschitz property of the base classifier and the variance of the Gaussian convolution which governs the induced level of smoothness." Yet, i don't see how this is a circular dependency.

* page 3: what do the authors mean by "furnished by $f$"?

* page 3: "concerning the $\ell_2$ norm" -> "w.r.t.... "

* page 3: In defining the margin of $f$ at $x$, it should be "of $f$ at $(x,y)$", since its definition uses the correct label $y$.

* Prop. 1: The first sentence is incomplete. Please revise English usage.

* The authors use the term "sub-classifier" to refer to the un-robustified function $f$, alas sometimes they refer to it as a "classifier" (as in Prop.3)

* It might help the reader to make explicit what the final classifier is (which I believe is $\argmax_{s,t} \tilde{F}^t_s(x)$, no?)

* In the conclusion: "In this paper, we demonstrate a significant correlation between the variance of randomized smoothing and two critical properties of the base classifier" I don't think the authors demonstrate any "correlation" between these things. Maybe they meant "connection"?

---

> ### Author Response · Authors · 2023-11-16
>
> We would like to thank the reviewer for their interest, positive remarks, and extensive comments.
>
> ## Weakness
>
> ### The paper is at times verbose, and at times unclear. Some comments are exaggerated or incorrect (See below).
> We have revised the paper in an effort to make it clearer.
>
> ### In my view, the contributions of employing a better concentration inequality to provide tighter tail guarantees for finite samples is small, as is the fact of changing the function that maps the sub-classifier predictions to the probability simplex.
>
> Our approach generalizes the usage of argmax as a simplex map and provides insight into why methods like (Salman et al. 2019) which reduce Lipschitz of $f$ are crucial.
>
> ## Question Major Points
>
> ### Throughout the paper, the authors refer to softmax (and other functions) as "projections" onto the probability simplex. The softmax is not a projection onto the simplex. In fact, this reviewer believes that the softmax is not a projection onto any set (is it?). Throughout the manuscript, the word "projection" is used in lieu of the term "function" or "map".
>
> We have modified the notation and replaced "simplex projection" by "simplex map".
>
> ### Randomized smoothing can provide a certified radius at varying levels, depending on the strength of the added noise. It is clear to me that leveraging the fact that $f$ is Lipschitz continuous allows the authors to find an optimal randomized smoothing strength, as indicated in Prop. 4. Such a smoothing strength results in a certified radius. What is unclear to me is, what can be done if one requires certification at a different radius?
>
> As we are doing Randomized Smoothing on an already Lipschitz classifier, we can as in Randomized Smoothing alone change the magnitude $\sigma$. It will not be optimal: the cross effect of Randomized Smoothing and Lipschtz Continuity will not be maximal, but it allows to reach for a potentially greater certified radius.
> We have also generalized our procedure for a simplex of mass $r$, changing $r$ allows to adapt to one targetted certified radius, as discussed in Section 3.4.
>
> ### page 3: The definition of $L(f)$ that the authors use require $f$ to be differentiable. This assumption is not necessary (as best as I can tell) for their results to hold (and indeed, not applicable to ReLU networks). So, just define $L(f)without assuming differentiability).
>
> We have modified the definition of $L(f)$ by:
>
> $L(f) = \sup_{\substack{x, x' \in \mathcal{X} \\ x \neq x'}} \frac{\lVert f(x) - f(x') \rVert_2}{\lVert x - x' \rVert_2} \enspace$.$
>
> ### The authors comment on some related works pertaining the analysis of Lipschitz-continuity for devising certified radii. There exist extensions of these ideas that refine the analysis to requiring only local Lipschitzness, as in [A], which lead to improvements on the certified radius. Can the authors comment on relations to that work, and on whether their results could apply in a local way as in [A] ?
>
> We have devised a Theorem to bound $||\nabla \Phi^{-1} \circ \tilde{f}_k (x) ||$. This theorem can be used to derive bound on a the local Lipschitz constant.
>
>
> ### page 4: the authors mention that "these bounds are relevant for the composite classifier defined as $\Phi^{-1} \circ f$. However, I believe they mean the classifier $\tilde{F}_{s, t}$, no?
>
> We have corrected the error, we meant the classifier $\Phi^{-1} \circ \tilde{f}_k$.
>
> ### he authors argue in Section 3.3 that different functions s(z) have different impacts in terms of the variance of s(Z) vs the variance in Z. This is quite obvious, and, respectfully, I find the Example 1 not useful/trivial. ii) It is unclear, however, how such a map should be chosen to be optimal. It seems to me (from their experimental section) that they simply compute different options based on different alternatives for s(z) and choose the one that results in larger empirical margin. Is this correct? Shouldn't one be able to provide a "best" choice for s(z)?
> For future works, I think, it is not obvious that there is a consensus. (60 \% is $\mathrm{argmax}$ rest $\mathrm{sparsemax}$).
> We are going to modify the procedure and add a phase where we determine $s$ and $t$ on a first $n_0$ sample of $\delta_i$, in the same manner as \cite{cohen_certified_2019}.
> Then run the standard, RS procedure with the classifier $F_{s, t}$ on another sample of size $n$.
> This new procedure is statistically correct.
>
> ### After Prop. 3, the authors mention that "This refinement on the bound was possible by supposing Lipschitz continuity on the base classifier $f$. Note that its Lipschitz constant can be arbitrarily high, so this assumption is quite light". What do the authors mean by "light" here?
>
> It means that the Lipschitz constant $L(f)$ can be arbitrarily high at the end we come up with a bound which does not depend on  $L(f)$. So it is not a restrictive hypothesis to assume and this conducts to a better bound for most networks used in Randomized Smoothing.

---

> ### Author Response · Authors · 2023-11-16
>
> ## Question Other Points:
>
> ### page 1: the authors write that there's a "circular dependency: the regularity of the smoothed classifier depends on the Lipschitz property of the base classifier and the variance of the Gaussian convolution which governs the induced level of smoothness." Yet, i don't see how this is a circular dependency.
>
> We have removed this formulation. It was used because the Lipschitz constant of the base classifier plays a dual role in the MC variance and the Lipschitz constant of the smoothed classifier.
>
> ### page 3: what do the authors mean by "furnished by $f$"?
> We reformulated this sentence by "given by $f$".
>
> ### page 3: "concerning the $\ell_2$ norm" -> "w.r.t.... "
> Same as above.
>
> ### page 3: In defining the margin of $f$ at $x$, it should be "of $f$ at $(x, y)$", since its definition uses the correct label $y$.
> We modified the definition of the margin accordingly.
>
> ### Prop. 1: The first sentence is incomplete. Please revise English usage.
> We rewrote the sentence.
>
> ### The authors use the term "sub-classifier" to refer to the un-robustified function $f$, alas sometimes they refer to it as a "classifier" (as in Prop.3)
> We corrected it.
>
> ### It might help the reader to make explicit what the final classifier is (which I believe is $\argmax_{s, t} \tilde{F}_s^t(x)$, no?)
>
> We corrected it and defined it clearly in the paper now.
>
> ### In the conclusion: "In this paper, we demonstrate a significant correlation between the variance of randomized smoothing and two critical properties of the base classifier" I don't think the authors demonstrate any "correlation" between these things. Maybe they meant "connection"?
>
> We corrected this term usage.

---

> > ### Comment · Reviewer_W4Xb · 2023-11-21
> > **Thanks for the clarification**
> >
> > I thank the authors for their comments: some have clarified some misunderstandings i had before, and other have corrected elements that were not initially completely correct in their paper.
> >
> > I'm increasing my score accordingly.

---

### Official Review · Reviewer_ay9y · 2023-10-31

**Soundness:** 3 good
**Presentation:** 2 fair
**Contribution:** 3 good
**Rating:** 8
**Confidence:** 3

**Summary:**

In this paper authors study the interplay between randomized smoothing, Lipschitz bounds and margin of the classifier, to design a new robustness certification method with improved guarantees.
More specifically, they show that when the Lipschitz constant of a base classifier $f$ is low, the robustness radius of the smoothed estimator $\tilde f$ can be lower, and the variance of the associated estimators is also lower. Therefore, with the same cost (in $n$ the number of samples) a bigger certified radius can be estimated at risk level $\alpha$.
The robustness radii rely on a margin, typically the difference between the two highest logits. Author show that `argmax` might not be the best option, and that other options like `sparsemax` have better properties.

Finally, the new certification method is tested and compared against randomized smoothing and Lipschitz-based certification, on Imagenet and Cifar-10 tasks.

**Strengths:**

### Originality

The work takes a new look at an existing topic, by incorporating Lipschitz constraints into randomized smoothing. It borrows tools from different fields, like concentration inequalities, projections onto the simplex (argmax, softmax, sparsemax), and robustness certification against adversarial attacks.

### Clarity

Fig 1. helps a lot to the understanding.

### Significance

The proposed algorithm LVM-RS can be seamlessly ingrated into existing frameworks, it is very general (with different temperatures and simplex projections), and not too expensive to run. The empirical results are convinving, since the new method improves upon both (i) Randomized Smoothing (ii) 1-Lipschitz networks of (Wang & Manchester, 2023).

**Weaknesses:**

I struggled a lot to understand the paper at times. There is a lack of details, and even with the help of the related literature (e.g. Cohen et al) I couldn't be sure of what the authors were trying to say. My questions are detailed below. Clarifications would benefit a lot to the paper.

**Questions:**

> Most of the randomized smoothing approaches simplify the margin $-\max_{y\neq k}\hat f_k(x)\geq -(1-\hat f_y(x))$ (Cohen et al., 2019)

Can you clarify where in the paper of Cohen? Are you referencing their choice $p_A=1-p_B$ in top of their p6 ?

> For each combination of (s, t), we determine the risk-corrected margin $M_2(p_{st} , \alpha)$.

Is it statistically correct (w.r.t $\alpha$) ? Since you perform multiple tests with the same scores $f(x+\delta_i)$ ?

### Sets and domains of functions

It is not always clear from context what are the definition of functions. For example:
* $\hat f$ is implicitly defined in top of page 4, I assumed you meant $\hat f_k()$ in the argmax. So $\hat f:\mathcal{X}\rightarrow\mathbb{R}^c$
* but $\Phi$ is a scalar function (per its definition), so what is the meaning of $\Phi^{-1}\circ \hat f$ ? Is it vectorized?
* so, is $g$ a scalar or a vectorized function Eq (3) ?

In Sec 2.1 the notation $L()$ is defined as a maximum of gradient norm over $\mathbb{R}^d$. But the gradient is only defined for scalar functions, not for multivariate functions (in which case it is a Jacobian, not a gradient). This makes sense to define $l(g)=\max_k L(f_k)$. But, then, if $g$ is multivariate, how can you write $L(g)$ ? What does it mean?

I suggest to define clearly each notation, and specify the domain of each function, not in an implicit manner in the middle of a formula.

### Runtime

What is the runtime cost of your method compared to randomized smoothing?

### Typo

The paper can be hard to read. In p6:

> Cohen et al. (2019); Salman et al. (2019); Carlini et al. (2023) use argmax simplex projection they can use the Clopper-Pearson Bernouilli tailored confidence interval ... etc

This paragraph should be rewritten, the first two sentences are not grammatically correct.

> Proposition 2

Is it a definition of the `shift` function in left hand side? Use `:=` notation like `\vcentcolon=` if so. Otherwise, define it somewhere. Please add a discussion of what $Z_i$ is supposed to be in *your* context.

> Once the variance has been minimized via Lipschitz constant regularization and that we can leverage
low resulting empirical variance with Empirical Bernstein’s inequality a vital aspect for achieving
a significant certified radius is negotiating the margins-variance trade-off

Add punctuation.

> This is because the compensatory shift, incorporated
to address the inherent risk, becomes minimal

What does this sentence mean?

> Equation (4)

$R_2$ and $R_3$ inequality should be reversed.

---

> ### Author Response · Authors · 2023-11-16
>
> We would like to thank the reviewer for their interest, positive remarks, and extensive comments.
>
> # Questions
>
> ### Most of the randomized smoothing approaches simplify the margin $-\max_{y \neq k} \hat{f}_k(x) \geq -(1 - \hat{f}_y(x))$ (Cohen et al., 2019) Can you clarify where in the paper of Cohen? Are you referencing their choice $p_A = 1 - p_B$ in top of their p6 ?
>
> Yes, it was the message we wanted to convey. In (Cohen et al.2019) they reduce the multi-class setting to a binary class setting, this reduction comes at the expense of a degraded certified radius.
> We have rewritten this part to make it clearer, please see Appendix C.
>
> ### For each combination of (s, t), we determine the risk-corrected margin $M_2(p_{s,t}, \alpha)$. Is it statistically correct (w.r.t ) $\alpha $? Since you perform multiple tests with the same scores $f(x + \delta_i)$?
>
> We have modified the procedure and added a phase where we determine $s^*$ and $t^*$ on a first $n_0$ sample of $\delta_i$, in the same manner as \cite{cohen_certified_2019}.
> Then run the standard, RS procedure with the classifier $F_{s^*, t^*}$ on another sample of size $n$.
>
> ## Set and domain of functions
> It is not always clear from context what are the definition of functions. For example:
>
> ### $\hat{f}$ is implicitly defined in top of page 4, I assumed you meant $\hat{f}_k$ in the argmax. So $\tilde{f} : \mathcal{X} \mapsto \mathbb{R}^c$
>
> We have modified this section and add a proper definition of the smooth sub classifier $\tilde{f}$.
>
> ### but $\Phi$ is a scalar function (per its definition), so what is the meaning of $\Phi^{-1} \circ \hat{f}$? Is it vectorized? so, is $g$ a scalar or a vectorized function Eq (3) ?
>
> We have modified the notation in the paper and only note $\Phi^{-1}(\tilde{f}_k(x))$ with $\Phi^{-1} : [0, 1] \mapsto \mathbb{R}$.
>
> ### I suggest to define clearly each notation, and specify the domain of each function, not in an implicit manner in the middle of a formula.
> We have done accordingly in the paper.
>
> ### In Sec 2.1 the notation $L()$is defined as a maximum of gradient norm over $\mathbb{R}^d$. But the gradient is only defined for scalar functions, not for multivariate functions (in which case it is a Jacobian, not a gradient). This makes sense to define $l(g) = max_k L(f_k)$. But, then, if $g$ is multivariate, how can you write L(g) ? What does it mean?
>
> We have modified the definition of the Lipschitz constant as $L(f) = \sup_{\substack{x, x' \in \mathcal{X} \\ x \neq x'}} \frac{\lVert f(x) - f(x') \rVert_2}{\lVert x - x' \rVert_2} \enspace$.
> We also remove the notation $l(f)$ and will use $L(f_k)$ instead.
>
> ### Runtime
> ### What is the runtime cost of your method compared to randomized smoothing?
> We have integrated runtime cost in the tables.
>
> ### Typo
> We have integrated your remarks.

---

### Official Review · Reviewer_12zu · 2023-10-31

**Soundness:** 2 fair
**Presentation:** 2 fair
**Contribution:** 3 good
**Rating:** 6
**Confidence:** 4

**Summary:**

This paper studies methods to obtain improved randomized-smoothing (RS) certificates against $\ell_2$ bounded adversarial attacks. The main contribution is to incorporate the lipschitz constant of the _base_ classifier $f$ into RS certificates to obtain improved bounds on the certified radius of the smoothed classifier $g$. Additionally, RS certificates hold with a probability $1 - \alpha$, where $\alpha$ is small. The auxiliary technical contribution of this paper is to utilize the variance of the output of $f$ to control $\alpha$ better than prior work. Empirical evidence supports the theoretical results, demonstrating improved certificates over prior work.

**Strengths:**

1. RS is currently the state of the art certification method for obtaining certificates of robustness against $\ell_p$ attacks. However, the interaction of the properties of the base classifier with the final certificate is not very well understood. This paper attempts an understanding of how the Lipschitz constant of the base classifier affects the final certificate — this is an interesting problem to study and is potentially very useful for the community.

2. Further, the paper revisits computation of the confidence $\alpha$ that the RS certificate hold with, and demonstrates how the variance of $f$ (the base classifier) can be used to improve $\alpha$, via a better concentration inequality. Exploring improved concentration inequalities for RS is also a useful research direction.

**Weaknesses:**

1. Writing: The writing of the paper is quite wordy in many places, and confusing at times, leading to major issues in readability and clarity. Details:

	-	Abstract: It is unclear what “variance introduced by RS”, “simplex projection”, “variance-margin tradeoff” mean. It is unclear what does bernstein’s inequality have to do with any of this.

	-	Introduction: Would be good to define prediction margin. All methods with certified radius deal with a way of controlling /estimating the Lipschitz constant of the network, it is unclear why is Tsuzuku et. al. specifically mentioned. The statements are also quite wordy here (“radius that encompasses … constant”), a simple $|f(x) - f(x + \delta)| \leq L ||\delta||$ would suffice here.
	-	Figure 1: What is radius binding, risk shift, simplex projection? “Risk” is used throughout the introduction without a definition.
	-	P2: The circular dependency mentioned in paragraph 2 was not clear, what exactly is the circular dependency?
	-	The definition of R(f, x) is written in a very convoluted fashion, isn’t it simply $\min_\epsilon$ such that $\exists$ adversarial example $\in B(x, \epsilon)$. The benefit of the written form of the definition is unclear.
	-	The correct earliest reference for Randomized Smoothing is not Cohen et al. 2019, please see the related work section in Cohen et al. 2019 for the correct references.
	-	Defining the certified radius for the composition of $\Phi^{-1}$ and $\tilde f$ is confusing, since the radius is to be used for certifying $\tilde f$, and not the composition.
	-	Section 2.4: Margins and Robustness — This is again too wordy “promise in bolstering classifier defense”, “inherent synergy between them”, “our exploration dives deep into this nexus unravelling novel insights …”. The message again simply seems to be that greater margin = larger certified radius (at fixed L), and could be stated in a single line.
	-	Section 3.1: Relation between certified Radii. The highlight of this section seems to be the last paragraph, that variance plays a crucial role in the RS certificates. It is unclear what does the rest of the section add to this message. Perhaps it would be better to simply stress this strong point, and detail Table 3.1.
	-	Table 3.1 is hard to parse, could simply plotting the difference between R2 and R3 convey the message better?

2. Technical Correctness: Related to the point above, there are several places where some additional steps / statements are needed to support the technical claims made in the paper:
	-	P3, Proposition 1. Isn’t the condition for correct prediction simply that margin > 2 * Lipschitz Constant * perturbation radius? ( Assume class 1 is majority at x, class 2 is second highest, then $|f1(x) - f1(x + v)| \leq L ||v|| = L \epsilon$, and $|f2(x) - f2(x + v)| \leq L epsilon$. Then, if $f1(x) - f2(x) > 2 L \epsilon$, then $f1(x + v) > f2(x + v)$). Why is the constant $\sqrt{2}$, instead of $2$?
	-	Appendix P12, Proof of Proposition 3.
		-	Why is $u \in R^c$ defined? Doesn’t seem to be used in the proof.
		-	Are there are several typos confusing $\delta$ with $\delta’$? For instance why is $E_\delta [v^\top \delta h(x + \delta)] = E_\delta [v^\top \delta’ h(x + \delta’)]$? The next step then replaces one of the $\delta$ with $\delta’$, which is unclear too.
		-	**Major 1**: Why is $v^T \nabla \tilde h(x) = (1/\sigma^2) E[…]$? Expanding $v^T \nabla \tilde h(x)$ gives $v^T \nabla \tilde h(x) = E[v^T \nabla h(x + \delta)]$, it is unclear how to proceed from here.
		-	In the step marked (i) in the proof, is there some typo? $|h(x + \delta) - h(x + \delta’)|$ is replaced by $\min(1, 2L) |v^\top \delta|$, why is this true? Isn’t this bound $L ||\delta - \delta'||$?
		-	**Major 2**: In the construction for the lower bound on $J$, $h_0 = (…) \min (…, l(h)…)$, but how can the constructed classifier depend on its own lipschitz constant. Some clarity is needed on why one can do this, and why this classifier exists.


3. Technical Clarity: There are several places where additional clarity would help the reader.
	-	It is unclear how one obtains (4) by using the bound mentioned in the paragraph preceding it. A few lines would be great for readers not familiar with the proof in Cohen et al. 2019.
	-	The Gaussian Poincare inequality seems to be very similar to other well known concentration inequalities for functions of gaussian random variables. A brief contextualization would be great. For instance, how does this relate to the standard concentration inequality for Lipschitz functions of sub gaussian random variables: $P( |h(Z) - E h(Z)| > \delta ) \leq \exp(-c \delta^2 / L(h))$ (for a reference, see Theorem 5.2.2 in High Dimensional Probability by Roman Vershynin)
	-	Eq (5), where is alpha in the RHS? Specifically, it is unclear how the $p_1, p_2$ have any explicit dependence on $\alpha$. I believe the dependence is that the number of samples n affects both alpha and $p_1, p_2$. But this is not clear at all by looking at (5).

	-	Related to the above point, now since the dependence was not specified in (5), it is unclear how $M_2$ is computed in the main algorithm Algorithm 1. Specifically, what is $\bar M_2(p, \alpha)$ on Line 6 of Algorithm 1, how does it relate to $s$, how does it relate to $t$?
	-	What are the simplex projections being used in line 4, algorithm 1? The text mentions “for various projections … spanning softmax and sparsemax” — Which projections specifically?
	-	Now, in order to compute the margin $M_2$, I believe one needs to know the lipschitz constant of the classifier $s$ composed with $f$, how is this computed for each simplex projection chosen?
	-	Experiments: How is the 1-Lipschitz backbone constructed. This should be detailed as it relates to the main contribution.
	-	Experiments: The effects of the change in the $\alpha$ calculation vs the change in the certificate are unclear. In particular, what are the results when the $\alpha$ calculation in this paper is used on top of Cohen et. al.? Similarly, what are the results when the Clopper-Pearson $\alpha$ calculation are used on top of this paper’s certificate?

**Questions:**

At a high level it would be good to,

(1) clarify and augment the paper with additional technical details to complete the proofs and correct the typos (details above),

(2) explain the precise mathematical details of the certification procedure from input to certificate (see details in weaknesses above), and

(2) empirically ablate the two components of the proposed approach (see details above).

---

> ### Author Response · Authors · 2023-11-16
>
> We would like to thank the reviewer for their interest, positive remarks, and extensive comments. We have revised the paper based on your comments to make it clearer. Please do not hesitate to look at the new version of the paper and let us know if there is still some misunderstanding.
>
> **1. Writing**
>
> **Abstract: It is unclear what “variance introduced by RS”, “simplex projection”, “variance-margin tradeoff” mean. It is unclear what does Bernstein’s inequality have to do with any of this.**
> We have modified the abstract to make these notions more clear.
>
> **Introduction: Would be good to define prediction margin. All methods with certified radius deal with a way of controlling /estimating the Lipschitz constant of the network, it is unclear why is Tsuzuku et. al. specifically mentioned. The statements are also quite wordy here (“radius that encompasses … constant”), a simple $| f(x) - f(x + \delta) | \leq L || \delta ||$ would suffice here.**
> We have included the definition of margin and Lipschitz in the introduction.
>
> **Figure 1: What is radius binding, risk shift, simplex projection? “Risk” is used throughout the introduction without a definition.**
> We have added a sentence in the introduction to define the risk: the probability that the MC classifier outputs the wrong answer. We have added a definition of the risk $\alpha$ in the related work section on randomized smoothing.
> We also have defined what is a simplex map (previously called simplex projection) in the introduction.
>
> **The circular dependency mentioned in paragraph 2 was not clear, what exactly is the circular dependency?**
> We have removed the use of this term. It was used to describe the different impacts of the Lipschitz constant of the base classifier on the MC variance and on the Lipschitz constant of the base classifier.
>
> **The definition of R(f, x) is written in a very convoluted fashion, isn’t it simply $\min_\epsilon$ such that $\exists$ adversarial example $\in B(x, \epsilon)$. The benefit of the written form of the definition is unclear.**
> We have modified the definition of $R(f, x)$ accordingly.
>
> **The correct earliest reference for Randomized Smoothing is not Cohen et al. 2019, please see the related work section in Cohen et al. 2019 for the correct references.**
> We have added the previous reference of Randomized Smoothing, Lecuyer 2019, Li 2019.
>
> **Defining the certified radius for the composition of and is confusing, since the radius is to be used for certifying $\Phi^{-1}$, and $\tilde{f}$, not the composition.**
> When performing Randomized Smoothing we estimate the smoothed classifier $\tilde{f}: R^d \mapsto [0, 1]$.
> With this estimation, we have by composing with $\Phi^{-1}$ the classifier $\Phi^{-1} \circ \tilde{f} : R^d \mapsto R$ which enjoys a better certified radius.
>
> **Section 2.4: Margins and Robustness — This is again too wordy “promise in bolstering classifier defense”, “inherent synergy between them”, “our exploration dives deep into this nexus unraveling novel insights …”. The message again simply seems to be that greater margin = larger certified radius (at fixed L), and could be stated in a single line.**
> We have reformulated those sentences to make them clearer, please see section 2.4.
>
> **Section 3.1: Relation between certified Radii. The highlight of this section seems to be the last paragraph, that variance plays a crucial role in the RS certificates. It is unclear what does the rest of the section adds to this message. Perhaps it would be better to simply stress this strong point, and detail Table 3.1.**
> We have put this section in the Appendix as it does not contain important insights for the rest of the contribution.
>
> **Table 3.1 is hard to parse, could simply plotting the difference between R2 and R3 convey the message better?**
> We have added a column with the difference between the radius.

---

> > ### Comment · Reviewer_12zu · 2023-11-21
> > **Re: Related Work**
> >
> > Thanks for incorporating the corrected references. I would like to point out a recent work studying other properties of the base classifier that affect the RS-classifier: [A] Pal et. al. (2023). Understanding Noise-Augmented Training for Randomized Smoothing. Transactions on Machine Learning Research.

---

> ### Author Response · Authors · 2023-11-16
>
> **2. Technical Correctness: Related to the point above, there are several places where some additional steps/statements are needed to support the technical claims made in the paper:**
>
> **Proposition 1**. Isn’t the condition for correct prediction simply that margin > 2 * Lipschitz Constant * perturbation radius? ( Assume class 1 is majority at x, class 2 is second highest, then $|f_1(x) - f_1(x + v)| \leq L ||v|| ) =L \epsilon$, and $|f_2(x) - f_2(x+v)| \leq L \epsilon$. Then, if $f_1(x) - f_2(x) > 2 L \epsilon$, then $f_1(x+v) > f_2(x +v)$. Why is the constant $\sqrt{2}$, instead of 2?
>
> The demonstration that you gave to certify $x$ for classifier $f$ is true when you assume $f_k$ to be $L$ Lipschitz continuous for every $k$. Here we have the assumption that $f$ is $L$ Lipschitz continuous, hence we can obtain a stronger certificate.
>
> Tsuzuku et al proposed a tighter version of this bound in Proposition 1 of [1].
>
> [1] Lipschitz-Margin Training: Scalable Certification of Perturbation Invariance for Deep Neural Networks

---

> ### Author Response · Authors · 2023-11-16
>
> ###  Appendix P12, Proof of Proposition 3.
>
> ### Why is $u \in \mathbb{R}^d$ defined? Doesn’t seem to be used in the proof
>
> It is a typo and has been corrected. It is used in the next proof for the bound on $L(\tilde{h}_k)$, but for $L(\tilde{h})$ it is not required.
>
>
> ### Are there are several typos confusing $\delta$ and $\delta^\prime$ For instance why is $E_\delta[v^\top \delta h(x + \delta)] = E_\delta[v^\top \delta^\prime h(x + \delta^\prime)]$  ? The next step then replaces one of the $\delta$ with  $\delta^\prime$, which is unclear too.
>
> $\delta$ is sampled from a ${\cal N}(0, \sigma^2 I)$ distribution. For any vector $v \in R^d$, such that $|| v ||_2 = 1$, we can write  $\delta = \delta^\perp + \tilde{\delta}$, where $\tilde{\delta} = (v^T\delta)v$ and $\delta^\perp \perp v$.
> Let $\delta' = \delta^\perp - \tilde{\delta}$.
> As $\delta^\prime \sim {\cal N}(0, \sigma^2 I)$ when $\delta \sim {\cal N}(0, \sigma^2 I)$
>
> $E_\delta[v^\top \delta h(x + \delta)] = E_\delta[v^\top \delta^\prime h(x + \delta^\prime)]$.
>
> Concerning the next step, we obtain that $v^\top \delta^\prime = v^\top (\delta^\perp - \tilde{\delta}) = 0 - v^\top \tilde{\delta}$, as $\delta^\perp$ is orthogonal to $v$.
>
> Thus
> $$ E_\delta[v^\top \delta h(x + \delta)] = - E_\delta[v^\top \delta h(x + \delta^\prime)]$$
>
> We can then write:
> $$
> E_\delta[v^\top \delta h(x + \delta)] = \frac{1}{2} (E_\delta[v^\top \delta h(x + \delta)] - E_\delta[v^\top \delta h(x + \delta^\prime)])
> $$
> we are doing this to use the Lipschitz property of $h$.
>
> ### Major 1 Why is $v^\top \nabla \tilde{h} (x) = \frac{1}{\sigma^2}E[...]$ Expanding $v^T \nabla \tilde{h} (x)$ gives $E[v^T \nabla h(x + \delta)]$ it is unclear how to proceed from here :
>
> We forgot to include Stein's Lemma [Charles M. Stein. Estimation of the mean of a multivariate normal distribution. The Annals of
> Statistics, 9(6):1135–1151, 1981] in the proof. It has been added.
>
> To compute $\nabla \tilde{h} (x)$ we use Stein's Lemma :
>
> Lemma:  (Stein's lemma) \\
> Let $\sigma > 0$, let $h : R^d \mapsto R$ be measurable, and let
> $\tilde{h}(x) = E_{\delta\sim {\cal N}(0, \sigma^2 I)}[h(x + \delta)]$. Then $\tilde{h}$ is differentiable, and moreover,
> $$
> \nabla \tilde{h}(x) = \frac{1}{\sigma^2} E_{\delta\sim {\cal N}(0, \sigma^2 I)}[ \delta h(x + \delta)] \ .
> $$
>
> We will add it to the proof.
>
>
> ### In the step marked (i) in the proof, is there some typo? $|h(x+\delta) - h(x+\delta^\prime)|$ is replaced by $\min(1, 2 L)|v^\top \delta|$}
>
> It is explained later in the proof :
>
> "where $(i)$ follows from the Lipschitz assumption on $h$ and the fact that $\|\delta-\delta'\| = \| 2\tilde{\delta} \| = \|  2(v^T\delta)v\| = 2|v^T\delta|$ and that $|h(x+\delta) - h(x+\delta')| \leq 1$"
>
>
> Using that $|h(x+\delta) - h(x+\delta^\prime)| \leq l(h) \|\delta-\delta'\|$ and what is mentioned above, we get $|h(x+\delta) - h(x+\delta^\prime)| \leq 2 l(h) |v^T\delta|$ and $|h(x+\delta) - h(x+\delta^\prime)| \leq 1$ as $h(x) \in [0, 1]$. Finally, it is inferior to the minimum of the two bounds: $|h(x+\delta) - h(x+\delta^\prime)| \leq \min(1, 2 l(h))|v^\top \delta|$.
>
> ### Major 2 In the construction for the lower bound on $J$. But how can the constructed classifier depend on its own Lipschitz constant? Some clarity is needed on why one can do this, and why this classifier exists.}
>
> There are several points:
> 1 - $h_o$ depends on the Lipschitz constant $l(h)$ of $h$.
> 2 - the Lipschitz constant of $h_o$ is $l(h)$
> 3 - $h_o$ is a piecewise linear function and its slope coefficient is determined by its Lipschitz constant.
> For instance, any affine function depends directly on its Lipschitz constant $x \mapsto lx + b$.
> Let us know if it sounds clearer to you.

---

> > ### Comment · Reviewer_12zu · 2023-11-21
> > **Re: Clarity in Proofs**
> >
> > Thanks for the clarifications. My concerns are mostly resolved, and I see the corrections in the paper. I think the proof techniques are novel, and am raising my score to reflect this.
> >
> > I am inclined to raise further if some lingering typos and minor edits are resolved: (1) $J$ has a supremum over $h$, so it is confusing to say $J(.., L(h_k))$. The correct notation would be $J(.., L) = \sup_{h \colon L(h) = L} ...$, i.e., the supremum is being taken over all $h$ with a specific lipschitz constant. (2) The previous point will also clear the confusion during the lower bound (3) In the lower bound, there is a typo $1$ to $r$ in the second equality (4) The lipschitz constant of $h_o$ indeed does seem to be $L$, but a small derivation should be included (e.g. for varying across any coordinate other than $x_1$, the function doesn't change, and $L$ is incurred only for $x_1$, there is however a kink at $r/2$ so please ensure that the lipschitz constant is ok there)

---

> ### Author Response · Authors · 2023-11-16
>
> ### 3. Technical Clarity : There are several places where additional clarity would help the reader.
>
> ### It is unclear how one obtains (4) by using the bound mentioned in the paragraph preceding it. A few lines would be great for readers not familiar with the proof in Cohen et al. 2019.
> This is a typo and has been corrected
>
> ### The Gaussian Poincare inequality seems to be very similar to other well known concentration inequalities for functions of gaussian random variables $P(|h(Z) - Eh(Z)| > \delta ) \leq \exp(-c \delta^2 / L(h))$. A brief contextualization would be great. For instance, how does this relate to the standard concentration inequality for Lipschitz functions of sub gaussian random variables: (for a reference, see Theorem 5.2.2 in High Dimensional Probability by Roman Vershynin)
>
> A discussion has been added to the paper in Section 3.2.
> The sub-Gaussian inequality involving the Lipschtiz constant of $s_k \circ f$,
> For $\delta_0, \delta_1, \dots, \delta_n$ i.i.d standard normal, for all $\mathrm{shift} > 0$:
> $$    \mathbb{P}[ \mathbb{E}[s_k(f(X + \delta_0))] - \frac{1}{n} \sum_{i=1}^n s_k(f(x + \delta_i)) \geq \mathrm{shift}] \leq \exp{\left( -\frac{\mathrm{shift}^2}{2 L(s_k \circ f)^2}\right)},
> $$
> The issue here is that computing $L(s_k \circ f)$ is NP-hard for common neural networks and the Lipschitz constant as a bound can overestimate the actual empirical variance.
> Most of the time the empirical variance provides a tighter bound on the true variance than $\sigma^2 L(s_k \circ f)^2$, which does not include the locality of the input $x$.
>
>
> ### Eq (5), where is alpha in the  RHS? Specifically, it is unclear how the  $p_1, p_2$ have any explicit dependence on $\alpha$. I believe the dependence is that the number of samples n affects both alpha and $p_1, p_2$. But this is not clear at all by looking at (5).
>
> In our framework the risk $\alpha$ is taken as a constant and for different sampling numbers it is the risk shift that is going to vary.
> We have made clear and apparent the dependency of $\alpha$ on $\bar{p}_1$ and $\bar{p}_2$. And on the risk corrected certified radius $R_2(\bar{p})$.

---

> ### Author Response · Authors · 2023-11-16
>
> ### What are the simplex projections being used in line 4, algorithm 1? The text mentions “for various projections … spanning softmax and sparsemax” — Which projections specifically?
> Projection are ranging $s \in \{ sparsemax, softmax, argmax \}$ and also ranging accross different temperatures from [0.01, 50]. Details has been added in the Section 3.5 and Section 4.
>
> ### Now, in order to compute the margin $M_2$, I believe one needs to know the lipschitz constant of the classifier composed with, how is this computed for each simplex projection chosen?
> To compute the certified radius of the smoothed classifier one needs indeed its Lipschitz constant. For an input $x$, and the smoothed classifier $\tilde{f}: \mathbb{R}^d \mapsto \mathbb{R}^c$ base on the sub classifier $f$, indeed, $\tilde{f}(x) := \mathbb{E}[s (f(x + \delta))]$. Let us call $p = \tilde{f}(x)$,  suppose that $p$ is sorted in decreasing order.
> The certified radius is given by $R_2(p) = \frac{\sigma}{2} (\Phi^{-1}(p_1) - \Phi^{-1}(p_2) )$. Here the Lipschitz constant of $\Phi^{-1} \circ \tilde{f}$ is $1/\sigma$ as derived by (Salman et al, 2019).
> There is no assumption needed on the Lipschitz constant of $s \circ f$. Therefore, for pure RS the choice of $s$ is not going to change the Lipschitz constant of $\tilde{f}$.
>
> ### Experiments: How is the 1-Lipschitz backbone constructed? This should be detailed as it relates to the main contribution.
>
> We have taken the Small Sandwich architecture as described, in (Wang & Manchester, 2023). We will add a section in the appendix describing with more detail the architecture and the by-design Lipschitz layer.
>
> ### Experiments: The effects of the change in the $\alpha$ calculation vs the change in the certificate are unclear. In particular, what are the results when the $\alpha$ calculation in this paper is used on top of Cohen et. al.? Similarly, what are the results when the Clopper-Pearson $\alpha$ calculation is used on top of this paper’s certificate?
>
> The Clopper-Pearson confidence interval can only be use for binomial samples, here we smooth scalar outputs. Therefore we can not directly compare the two.
> We have included an ablation study to measure the impact of Bernstein's inequality over Hoeffding's inequality (previously used in (Lecuyer et al. 2019, Levine et al. 2020).
>
> ## Questions
> At a high level, it would be good to,
>
> ### (1) clarify and augment the paper with additional technical details to complete the proofs and correct the typos (details above),
>
> We have added complementary details to the proof.
>
> ### (2) explain the precise mathematical details of the certification procedure from input to certificate (see details in weaknesses above), and
>
> We have completed the algorithm describing the LVM-RS procedure, see Algo. 1.
>
> ### (2) empirically ablate the two components of the proposed approach (see details above).
>
> We have added Fig.4  to compare Hoeffding's inequality previously used in (Lecuyer et al,2019) and (Levine et al, 2020) with Bernstein's inequality and their impact on risk corrected margin.
> We have not included the Clopper-Pearson interval as it can only be used on Bernoulli trials and not on scalar outputs.
>
> Sparsemax, Argmax, and Softmax and their impact on margin are compared in Fig.5.

---

> ### Author Response · Authors · 2023-11-21
> **Update in Proofs**
>
> Thank you for your reply and for your involvement in reviewing our paper.
> Following your suggestion, we have changed $J(\sigma, L(h_k))$ to $J(\sigma, L)$ where $L$ is the Lipschitz constant of the subclassifier being smoothed. We have corrected the typo in (4) and provided a lemma to justify why the example $h$ used in the lower bound is indeed $L$-Lipschitz. Please see the updates in the proofs in Appendix C.
> We have also included your suggested reference to the effect of noisy training on the subclassifier over the smoothed classifier in Section 3.1.

---

### Official Review · Reviewer_2wVc · 2023-11-02

**Soundness:** 3 good
**Presentation:** 3 good
**Contribution:** 3 good
**Rating:** 6
**Confidence:** 3

**Summary:**

This paper identifies a circular dependency between the regularity of the smoothed classifier, the Lipschitz of the base classifier and the variance of gaussian noise in randomized smoothing procedure. Then they propose the LVM-RS framework, given either a Lipschitz constant or variance, one can select the complementary variance or Lipschitiz constant to maximize the synergistic effect of RS and Lipschitz continuity. Experiments show superior performance of the proposed method comparing with RS or deterministic Lipschitz training alone.

**Strengths:**

This paper identifies an interesting circular dependency among each ingredient in the RS procedure, and propose a principled way to improve current certification radius in a zero-shot manner.

In general, the paper is well written and each statement seems to be well supported by proof, empirical evaluation and intuitive explanation. Especially Figure 1 did a great job in summarizing the contribution of this paper.

Experiment results shows superior performance to the current state-of-the-art methods.

**Weaknesses:**

1. In terms of presentation, I feel the authors could add an overview paragraph for section 3. Sometimes I fail to connect each sub-sections. It may be good to map the flow of section 3 to Figure 1.

2. It will be good to discuss the computational cost of LVM-RS comparing to RS and deterministic Lipschitz.

**Questions:**

1. Is there a typo in Eq 4 where \leq should be \geq? Currently it contradicts table 1.

2. In algorithm 1, how many temperature do we need? And how to find those temperature?

3. How much computational cost that is induced by using multiple temperature? How sensitive the certified accuracy is with respect to the corrected margin?

---

> ### Author Response · Authors · 2023-11-16
>
> We want to thank the reviewer for their interest, positive remarks, and comments.
>
> **Weakness 1: In terms of presentation, I feel the authors could add an overview paragraph for section 3. Sometimes I fail to connect each sub-section. It may be good to map the flow of section 3 to Figure 1.**
> Section 3.1 has been removed from the body of the paper as it was not very insightful to the core message of the work.
>
> **Weakness 2: It will be good to discuss the computational cost of LVM-RS compared to RS and deterministic Lipschitz.**
> It has been added to the paper in the presentation of the LVM-RS procedure. Time computational costs have been added to the tables.
>
> **Question 1: Is there a typo in Eq 4 where \leq should be \geq? Currently, it contradicts Table 1.**
> We corrected the typo in Eq. 4., it has been put in Appendix.
>
> **Question 2: In algorithm 1, how many temperatures do we need? And how to find those temperatures?**
> In Algo. 1 we use 50 temperatures ranging from $0.1$ to $50$.
> Temperatures are chosen with a grid search. This has been added to the paper, in the description of the experiments.
>
> **Question 3: How much computational cost is induced by using multiple temperatures?**
> Using multiple temperatures is going to induce several application of simplex mappings, but the inference in the model and the sampling is done once. Hence it is not the bulk of the computation.
> We included the time of computation for RS, LVM-RS, and Lipschitz deterministic methods.
>
> **How sensitive the certified accuracy is with respect to the corrected margin?**
> Certified accuracy accounts for the frequency of input in which the certified radius is above a threshold $\epsilon$, the certified radius depends linearly on the margin.
> Can you please specify your question?

---

> > ### Comment · Reviewer_2wVc · 2023-11-23
> >
> > Thank you for answering my questions. Reorganizing section 3 has made the message clearer in the paper. For my last question, I wanted to see the certified accuracy curve with respect to the temperature (Fig 5 on page 24). However, it is a minor thing and overall I am ok with accepting the paper.

---

### Author Response · Authors · 2023-11-16
**General comment**

We thank a lot all the reviewers for providing high-quality feedback. Your insightful comments will significantly contribute to enhancing the quality of our paper. We have incorporated all the suggested considerations into a revised version of the paper. Should there be any remaining uncertainties or if further clarification is needed, please do not hesitate to bring it to our attention.

We have removed the old section 3.1 as it does not provide crucial insight to the paper. The content has been distributed in the related work section on Randomized Smoothing.

We also have extended some of our results for a simplex map on a simplex that sums to $r$.

---

### Comment · Reviewer_ay9y · 2023-11-17
**Thank you for your detailed answers**

The quality of writing has improved significantly in the revised version of the paper. The messages conveyed by the paper are more clear.

The experimental results show a significant advantage of the new method, with proper theoretical motivation.

To the best of my knowledge, this is the first time that Lipschitz constraints and randomized smoothing approaches are combined. This bridge is important for the community interested in robustness certification.

I improved my score accordingly.

---

### Public Comment · ~Vaclav_Voracek1 · 2023-11-25
**General comment**

Dear Authors, please, consider the following comments for the final version.

* I could not find what is the classifier (i.e., how to perform inference) and what property is being certified during the LVM-RS certification; Please, make this connection clear and show how does the certification procedure translate into guarantees.

* It would also be great to see an experiment demonstrating how the method improves over the standard R2 certification (= with argmax simplex map).

* I personally would not claim the introduction of empirical Bernstein bound as one of main contributions, as it was suggested already in the very first RS paper (Lecuyer, 2018).

Best,
Vaclav

---

### Meta-Review · Area_Chair_mUvn · 2023-12-15

**Metareview:**

This paper studies randomized smoothing and analyses the Lipschitz-variance-margin tradeoff. Four experts evaluated the work and found it valuable and interesting. The reviewer provided a large number of comments on the presentation of the results and the technical development. However, the authors have satisfactorily addressed the concerns and improved the quality of the paper to streamline the presentation and highlight the contribution.

**Justification For Why Not Higher Score:**

I reached this decision by evaluating the contributions and novelty of the work, taking into consideration both the reviews and the responses from the authors.

**Justification For Why Not Lower Score:**

I reached this decision by evaluating the contributions and novelty of the work, taking into consideration both the reviews and the responses from the authors.

---

### Decision · Program_Chairs · 2024-01-16

Accept (poster)